# Predicting metabolic adaptation from networks of mutational paths

Christos Josephides[1] & Peter S. Swain [ID] [1]

Competition for substrates is a ubiquitous selection pressure faced by microbes, yet intracellular trade-offs can prevent cells from metabolizing every type of available substrate. Adaptive evolution is constrained by these trade-offs, but their consequences for the repeatability and predictability of evolution are unclear. Here we develop an eco-evolutionary model with a metabolic trade-off to generate networks of mutational paths in microbial communities and show that these networks have descriptive and predictive information about the evolution of microbial communities. We find that long-term outcomes, including community collapse, diversity, and cycling, have characteristic evolutionary dynamics that determine the entropy, or repeatability, of mutational paths. Although reliable prediction of evolutionary outcomes from environmental conditions is difficult, graph-theoretic properties of the mutational networks enable accurate prediction even from incomplete observations. In conclusion, we present a novel methodology for analyzing adaptive evolution and report that the dynamics of adaptation are a key variable for predictive success.

[1] SynthSys—Synthetic and Systems Biology, School of Biological Sciences, University of Edinburgh, Edinburgh EH9 3BF, UK. Correspondence and requests for materials should be addressed to P.S.S. (email: peter.swain@ed.ac.uk)

Elucidating the factors that influence the emergence, diversity, and stability of microbial communities is a central interest in both ecology and evolution[1]. To predict and control community structure and function, it is necessary to understand how interactions between microbes and the environment manifest as selective pressures driving microbial adaptation and diversification[2].

Microbes frequently grow on mixtures of metabolic resources where competition for these growth-limiting substrates is a ubiquitous selection pressure. Surveys have revealed that microbes do not simultaneously use all available substrates. Instead, each species in a community specializes to a few substrates[3, 4]. This observation is anticipated by ecological (resource-ratio) theory, which posits that constraints in the capacity of an organism to use multiple substrates are necessary for diversifying selection in homogeneous environments[5], and is supported by competition experiments where trade-offs in using one substrate over another maintain metabolic polymorphisms[6–10].

The conditions under which a microbial population can invade another, and whether a stable community can be formed, are the subjects of ecological invasion analysis. These analyses typically either assume competition between infinitesimally-varying phenotypes[11, 12] or are not concerned with the mutational paths[13, 14] that may be generated through successive mutations and invasions. Advances in experimental evolution, however, now enable tracking of microbial lineages for hundreds of generations[15, 16] and expose how sequences of mutations shape the evolutionary process through competition between possibly disparate phenotypes[8]. Moreover, these experiments reveal evolutionary trajectories with both parallel and unique dynamics[17, 18], as well as variability in long-term outcomes[10].

To investigate how the interplay between metabolic constraints, environmental conditions, and the distribution of mutations influences the adaptation process, we developed a model that combines microbial chemostat ecology with an evolutionary process. Microbes compete at the ecological level for two substrates that can be growth-limiting, and we incorporate a phenomenological, metabolic trade-off between the consumption of one substrate relative to the other. Cells inherit this phenotype through nearly-faithful (clonal) reproduction, but rare mutations can change a cell's degree of specialization for the substrates. We do not restrict the size of the effect of mutations so that, in the extreme case, any phenotype can mutate to any other. Relaxing the common assumption of small mutations implies that standard methods from evolutionary invasion analysis[11, 12] are insufficient to describe the full repertoire of adaptation dynamics. We develop a new methodology with which we determine and analyze all mutational paths over a finite set of phenotypes and describe both transient and long-term evolutionary behavior.

We begin by introducing the model of chemostat ecology, which we then embed in a Markov process of mutation-limited adaptation to complete the eco-evolutionary model. We show that evolutionary adaptation that is generated by rare mutations can be visualized as a network of mutational paths connecting the states of the microbial community. We next analyze the Markov process to determine the possible long-term behaviors and show that multiple evolutionary outcomes emerge, including quasi-periodic outcomes and outcomes with more than one exclusive stationary state. We show that these outcomes are not restricted to distinct environmental conditions and that small environmental perturbations can change one type of outcome to another. In spite of this sensitivity to environmental conditions, the processes that lead to each evolutionary outcome have characteristic dynamics of adaptation, which are reflected in graph-theoretic properties of the mutational paths. Finally, we show that evolutionary outcomes can be predicted using these network properties

from incomplete observations of evolving microbial communities —even without knowledge of environmental conditions.

## Results

**Microbial ecology in the chemostat with two substrates**. Microbial evolution is often studied in chemostats, which maintain long-term cellular growth at a rate fixed by the experimenter, and we focus on a model of microbial ecology suitable for chemostats[19, 20]. The chemostat's environment is controlled through the choice of metabolic substrates, and to investigate microbial metabolic adaptation, we extend the standard model[21] to include two growth-limiting substrates, which we name $u$ and $v$ (Fig. 1a). We follow the typical assumption that all other necessary nutrients are provided in excess and do not limit growth. Both substrates are added continuously to the chemostat at influx rates that the experimentalist determines and microbes grow in proportion to the substrate concentrations. The two substrates are individually sufficient for growth—i.e., they are perfectly substitutable[22]. The chemostat is continuously diluted at a constant rate, which is the same for both microbes and substrates. Therefore, microbial populations must grow at least as fast as the dilution rate to survive being washed out (to extinction), and population growth exactly equals the dilution rate at steady-state. The chemostat is well-mixed through stirring, which eliminates the spatial component of the environment and facilitates the development of simpler mathematical models (Methods section).

In total, the ecological model is described by nine environmental parameters: the chemostat's dilution rate; the influx rates of the two substrates into the chemostat; the two maximal rates of cellular import for the substrates; the metabolic rates for each substrate once imported into a cell; and the yield of each substrate (an integer number of incremented growth states per unit of substrate metabolized).

**A cellular trade-off to constrain metabolic specialization**. Cells in the chemostat encounter both $u$ and $v$ substrates but cannot simultaneously specialize to using both. To investigate how a metabolic constraint affects adaptation[23], we consider a simple phenomenological trade-off between the ability of a microbe to import and consume $u$ substrate molecules compared to its ability to import and consume $v$ substrate molecules from the environment. We parameterize the metabolic specialization of each cell —its phenotype—by a number, $s$, between zero and one: values near zero indicate that the cell specializes to $v$; values near one indicate $u$ specialization (Fig. 1). Cells inherit their phenotype (their value of $s$) from their parent nearly always without variation through clonal replication and we assume that cells cannot spontaneously change their phenotype (Fig. 1b).

We can interpret the metabolic specialization, $s$, in two ways. In the first interpretation, cells can always import both substrates, but increase the rate of import for $u$ at the expense of decreasing the import rate for $v$. The maximal rate of import for $u$ is multiplied by $s$; the maximal rate of import for substrate $v$ is multiplied by $(1 - s)$ (Supplementary Note 1). Such a constraint may arise if, for instance, a finite intracellular resource is shared between the production of the $u$ and $v$ permeases[24–26] or through antagonistic pleiotropy[6, 7, 27]. In the second interpretation, cells adopt a mixed strategy (following evolutionary game theory[28]) of randomly switching between two metabolic programmes, each exclusive to only one substrate. Then, the probability of adopting the $u$ metabolic program is $s$ and the probability of adopting the $v$ program is $(1 - s)$ (Supplementary Note 2).

We model the microbial life cycle through a sequence of import and metabolism of substrates and a corresponding growth in biomass. Cells import the $u$ and $v$ substrates at rates

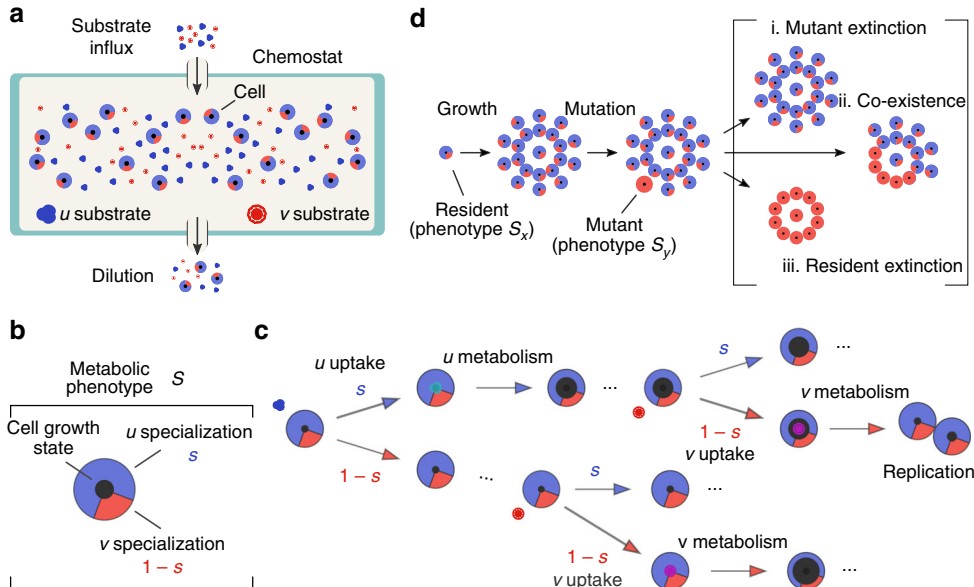

**Fig. 1** Modeling microbial ecology in the chemostat with a metabolic trade-off between two substrates. **a** The chemostat is a homogeneous environment. The two substrates ($u$ and $v$) are added continuously, and cells and substrates are diluted at a constant rate. **b** Cells specializing in using one substrate cannot also specialize in the other. The metabolic specialization of each cell is parameterized by a number, $s$, between zero and one: values near zero indicate a $v$-specialist; values near one indicate $u$-specialist; intermediate values are generalists. **c** To replicate, cells must progress through a series of discrete states of growth by metabolizing substrates. The yield of a substrate is the amount by which a cell's state of growth increases by metabolizing the substrate. Cells replicate when their growth state exceeds a critical value. When a cell encounters a $u$ substrate molecule, the molecule is imported with probability of $s$; when a cell encounters a $v$ substrate molecule, the molecule is imported with a probability of $(1 - s)$. **d** A rare mutation leads to competition between the (high abundance) resident phenotype and the (low abundance) invading phenotype and has three possible qualitative outcomes

proportional to the concentrations of these substrates in the chemostat, and the imported substrates are metabolized to give biomass to the cell, which divides once a threshold of biomass is crossed. We therefore structure the microbial population into discrete states of growth through which the cell progresses before replicating. Accordingly, cells that metabolize $u$ and $v$ molecules transition to a higher state of growth by an integer amount of states, which we call the substrate's yield, and replicate when they exceed a maximum growth state (Fig. 1c).

**Mutation-limited adaptation as a Markov process**. In our model, microbes will almost always produce offspring that inherit their parents' metabolic specialization, but, rarely, a mutation may result in a phenotypically distinct population that will compete with the (resident) parent population or community. We follow the theory of adaptive dynamics to include phenotypic mutations only on evolutionary timescales[11, 12]; i.e., mutations are sufficiently rare that mutants emerge only after the chemostat's ecology has reached steady-state. This assumption is typically referred to as the 'weak mutation' limit[29] and separates the ecological and evolutionary timescales. If, for example, the chemostat contains a single phenotypic population then invasion by a mutant has three possible outcomes: the mutant becomes extinct, or the resident becomes extinct, or the two phenotypes co-exist in a community (Fig. 1d). In our spatially homogeneous model at most two populations can co-exist—following the competitive exclusion principle[30], which states that at most $k$ species may co-exist on $k$ growth-limiting substrates.

**Simulating the map of invasion events**. When adaptation is limited by the availability of mutations, mutational paths emerge according to the sequence and outcome of mutation and invasion events. Successful invasion of a resident community by a new mutant modifies the chemostat's environment through changing

the steady-state levels of the available substrates. Therefore the context in which future mutation and invasion events occur is modified through the construction and destruction of ecological niches[2, 8, 18, 23, 31].

To generate the mutational paths, we first create an invasion map for the outcome of competitions between all resident communities and all mutants. To do so efficiently, we developed a dynamic programming algorithm to simulate invasion events assuming rare mutations on a discretized phenotype space (Fig. 2a). Briefly, the algorithm treats the invasion map as a tree: nodes are communities of phenotypes that can co-exist at steady-state and are connected by edges representing single mutation and invasion events. The algorithm iteratively constructs the tree by perturbing the steady-state of the community at each parent node to introduce a small population of mutants. The resulting competition is simulated (Methods section), and the outcome at the new steady-state is analyzed and recorded as a connected child node. We terminate a path of mutation and invasion events if the path arrives at a node that has already been placed in the tree. By doing so, we avoid redundant simulations because the sub-trees below two identical nodes are always the same. The algorithm completes when all paths have terminated (Supplementary Note 3).

We found that invasion fitness—the ability of a phenotype to invade another when initially rare—was dependent on the frequency and type of other phenotypes in the environment. For example, in the example shown in Fig. 2a, phenotype $C$ can always invade and drive $A$ to extinction in pairwise competition; the converse is not possible. Phenotypes $B$ and $C$ are mutually invasible and establish a community of co-residents. This community can be invaded by phenotype $A$, driving $B$ to extinction, thereby establishing a co-existence with phenotype $C$. The $A,C$ community would not have been possible without the intermediate environmental modification effected by phenotype $B$. Frequency-dependent effects through niche creation and

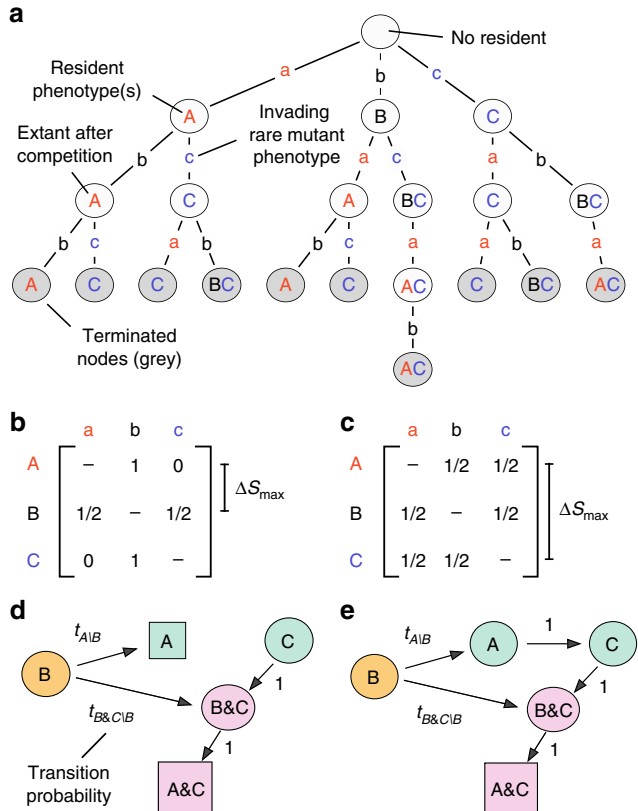

**Fig. 2** Generating a Markov process for mutation-limited adaptation. The procedure has two stages: first, simulating all possible ecological invasions; second, calculating the probabilities to transition between steady-states as a result of these invasions. We demonstrate a simple example that has only three possible phenotypes (labeled *A*, *B*, and *C*). **a** The invasion map contains the outcome of all possible competitions between residents and mutants and all steady-states. An algorithm based on dynamic programming constructs the invasion map as a tree: nodes represent steady-states connected via edges that represent mutations and competitions. Starting with the empty chemostat (root node), each state is perturbed with a small population of mutants that have each phenotypic value in turn. The steady-state after every competition is recorded in a child node and connected to the parent node. The algorithm dynamically truncates the tree to avoid any recursion (repetition). **b** To calculate transition probabilities between resident states, we parse the invasion map in **a** with a mutational process. We assume that mutations are uniformly distributed in the space of phenotypes, but centered on the resident's phenotype and with a maximum size of mutation $\Delta S_{max}$. The elements of the matrix show the conditional probability that phenotype $s_x$ generates mutant phenotype $s_y$. **c** The Markov process in **b** visualized through a graph of its directed network. States are classified as either transient (*circles*) or recurrent (*squares*). **d, e** As in **b, c**, but with a larger maximum size of mutation $\Delta S_{max}$. The recurrent states, as well as the mutational paths of the process, depend on the distribution of mutations

destruction are stronger when mutations in metabolic specialization are large because mutants can exploit metabolically disparate niches.

The invasion map can be thought of as providing two extensions to the pairwise invasibility plots used in adaptive dynamics theory[11, 12, 32]. Like adaptive dynamics, our modeling approach recovers the existence of evolutionary branching points where an initially monomorphic population branches into two co-existing populations (Supplementary Fig. 1), but unlike adaptive dynamics such points require no special treatment. In

addition, we can investigate non-local evolutionary branching that occurs after a mutation of large effect and analyze mutational paths that contain this type of diversification (Supplementary Fig. 2).

**Mutations connect community states**. To model adaptive evolution as a Markov process, we define a state of the Markov chain as a viable population in the chemostat, which is either monomorphic with all cells having the same phenotype or dimorphic (i.e., a community) with two distinct phenotypes that co-exist, and require the probability of a transition from one state to another through a single mutation and invasion event (Methods section). The probability of such a transition is proportional to the abundance of cells in the source state and to the propensity with which each cell generates the mutation that effects the transition to the target state. We will only be concerned with the sequence of transitions between one state to another and not with the waiting time between such events. As a consequence, our approach draws no conclusions about the length of time a microbial community spends in a particular state and hence does not quantify the evolutionary timescale. It should be noted, however, that in an experimental setting we may be able to observe (or reconstruct) transitions between states without necessarily knowing the precise time at which the transitions occurred. Under these circumstances, our methods, which work without a temporal component, retain their descriptive and predictive utility.

The availability of phenotypes to mutants partly determines which of the possible states will follow a current state. To investigate how the supply of mutations affects adaptation we use a uniform distribution of mutations in phenotype space centered on a resident phenotype, $s_x$, with a maximum mutation of size $\Delta S_{max}$, which is the largest possible difference between the metabolic specialization of a parent and that of its offspring. Mathematically, a mutant with phenotype $s_y$ can be generated by a cell with phenotype $s_x$ if $|s_y - s_x| \leq \Delta S_{max}$. For example, consider the $s = 0.75$ phenotype, which uses more $u$ substrate than $v$ substrate. When $\Delta S_{max} = 0.25$, cells with this phenotype can generate pure $u$ specialists ($s = 1$) but not pure $v$ specialists ($s = 0$) because a larger phenotypic change is required to achieve the latter. The probabilities of mutation are chosen so that all of the discrete $s_y$ values that can be generated from $s_x$ are equally likely after correcting for boundary effects (Fig. 2b, c). The distribution of mutations affects both the mutational paths and the stationary behavior of the adaptation process, and we parse the invasion map with different sizes of maximum mutation to investigate this dependency. For simplicity, we assume that the rate of mutation does not depend on the phenotype and that it is constant over time (Methods section).

Finally, to complete the description of mutation-limited adaptation as a Markov process, we must choose an initial distribution of ancestral states. Most laboratory evolution experiments start with a single isogenic population, and in our model adaptation begins with equal probability from any viable population comprising a single phenotype (rather than a community with two phenotypes).

To determine the long-term evolutionary outcome of an adaptation process, we classify the states of the Markov process as either transient or recurrent, following the theory of Markov processes. Transient states represent microbial communities that may only be visited once on a mutational path. A recurrent state, however, will always re-emerge once established, and the endpoints of mutational paths are always recurrent states (Fig. 2d, e). Recurrence occurs either when a microbial community cannot be invaded by any mutant that can be

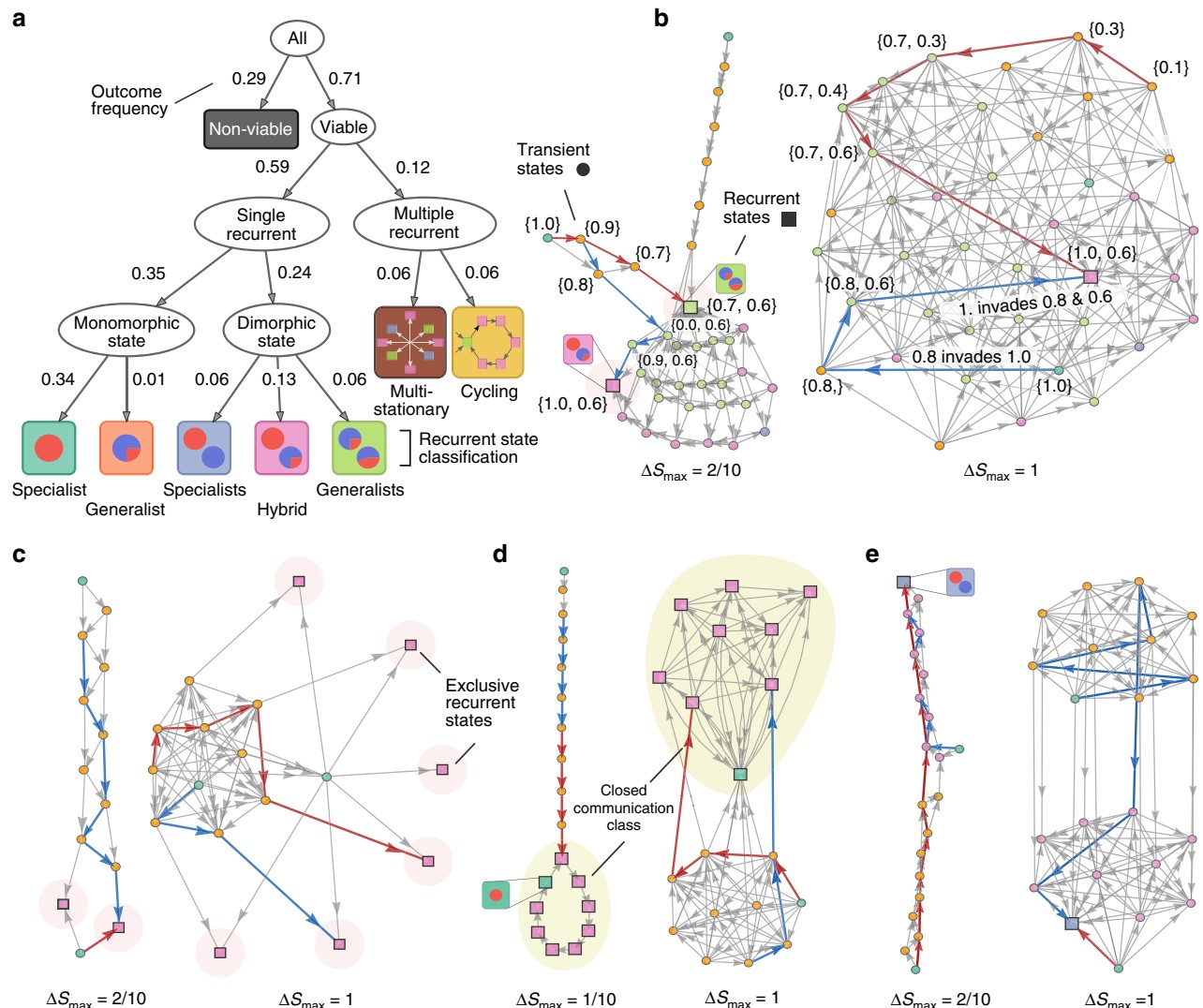

**Fig. 3** Our model generates multiple evolutionary outcomes and complex networks of mutational paths. Qualitative properties of the dynamics of adaptation can be visualized and conveyed through graphing the networks of mutational paths. **a** We hierarchically classify adaptation processes according to their evolutionary outcome using the number and type of recurrent states. Labels show each outcome's frequency obtained by sampling of environmental parameters. **b**–**e** The networks of mutational paths from four sets of environmental parameters demonstrate the scope of the model's dynamic and stationary behavior. The maximum size of mutation ($\Delta S_{max}$) affects the mutational paths and long-term outcomes of adaptation (compare the recurrent states between the *left* and *right* networks). *Circles* are transient states; *squares* are recurrent states; *colors* denote the number, and type of metabolic phenotypes in each state following **a**. *Red and blue traces* are examples of mutational paths, which start from an ancestral initial state (always a monomorphic state) and end at a recurrent state. We indicate the phenotype values of residents in some vertices to aid interpretation. **b** The invasibility relationship between a pair of phenotypes may reverse in the presence of co-residents as a consequence of frequency-dependent modification of the environment. For example, an $s == 0.8$ metabolic generalist can invade an $s = 1.0$ specialist but the converse is not possible in pairwise competition. When an $s = 0.6$ generalist is a co-resident, however, the $s = 1.0$ specialist can invade the dimorphic community of $s = 0.8$ and $s = 0.6$ generalists——and drive $s = 0.8$ to extinction. **c** A process with multiple, non-connected recurrent states has more than one evolutionarily stable state, each of which is reached with some probability. **d** A process where multiple recurrent states are connected exhibits quasi-periodic evolutionary cycling. **e** An example of a potentially bottle-necked process. The network consists of two highly-connected (*top* and *bottom*) subgraphs, which are themselves connected via only a few mutation and invasion transitions

generated from that community (an evolutionarily stable state[28]) or if there is a sequence of mutation and invasion events that returns to the community.

**Hierarchy of evolutionary outcomes**. To investigate the effect of environmental and mutational parameters on the adaptation process, we randomly sampled 10,000 sets of our model's nine environmental parameters. For each set of parameters, we calculated the invasion map using 11 discrete phenotypic values

ranging from a pure $v$-specialist to a pure $u$-specialist: $s \in \{0.0, 0.1,\ldots, 0.9, 1.0\}$. We parsed each invasion map with distributions of mutations that are uniform but with an increasing size of maximum mutation, $\Delta S_{max} \in \{0.1, 0.2,\ldots, 0.9, 1.0\}$, to generate 10 adaptation processes. We then analyzed the 100,000 resulting adaptation processes to determine their long-term evolutionary behavior.

We classify the evolutionary outcome of an adaptation process by the number and type of its recurrent states (Fig. 3a). At a high level, we differentiate between processes with either a single

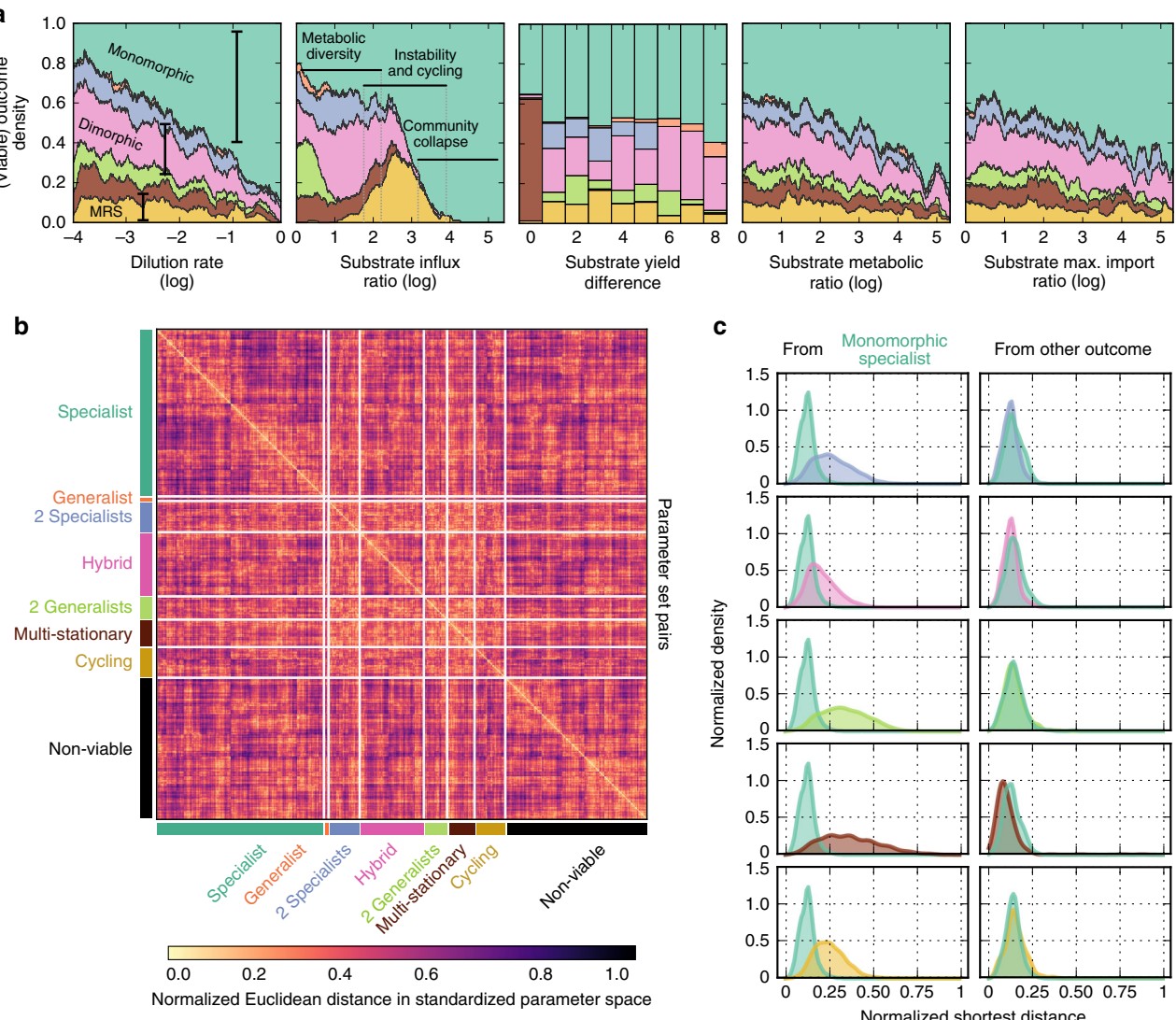

**Fig. 4** A complex association exists between the environment and the evolutionary outcomes. Overlapping clusters of evolutionary outcomes in environmental parameter space make prediction difficult and explain sensitivity of microbial communities to environmental change. **a** Some evolutionary outcomes are more likely to be associated with certain parameter values, but without a simple dependence. Using a nearest-neighbors algorithm, we empirically estimated the density of each outcome from sampling parameters. For all combinations of parameters except the dilution rate, we show densities at absolute values because of the symmetry about zero. Colors for the outcomes follow Fig. 3a. **b** Evolutionary outcomes form clusters in the space of parameters but these clusters are not distinct. We used Euclidean distance in a standardized space of parameters to perform within-outcome hierarchical clustering and then aligned clusters with samples from other outcomes to assess their overlap. We only plot a representative sample (1%) of the data set, which preserves the observed frequencies of each outcome. Distinct clusters would appear as bright regions in the diagonal combined with dark regions in the off-diagonal rows and columns indicating that samples with the same outcome are closer to each other than they are to samples with different outcomes. Instead, we found that clusters for one outcome are almost always proximal to samples from other outcomes (bright off-diagonal regions). **c** Environmental perturbations often cause community collapse and only rarely create diverse communities. We calculated the distance in the standardized space of parameters from each process to the closest process in all other outcome classes and so formed distributions of shortest distances. The left column shows per panel the shortest distances from the monomorphic specialists to themselves and to one other outcome. The monomorphic specialists are generally closer to other monomorphic specialists than they are to other outcomes. The right column shows per panel the shortest distances from one other outcome to itself and to the monomorphic specialists

recurrent state or multiple recurrent states. We further classify phenotypes as either specialists (when $s = 0$, a $v$-specialist, or $s = 1$, a $u$-specialist) or generalists (when $0 < s < 1$) and determine whether recurrent communities are either all specialists, all generalists, or a mixture of both. We observed two qualitatively different behaviors in processes with multiple recurrent states, which we named 'multi-stationary' and 'cycling'.

Multi-stationary outcomes have more than one exclusive recurrent state and each recurrent state is reached with some probability. In these adaptation processes we expect experimental replicates to eventually show divergent phenotypic variability[8, 10, 33]. We found that the number of recurrent states and their stationary probability can depend on the maximum size of mutation (Fig. 3b, c). Alternative microbial community states have been detected in the human gastrointestinal tract[34], although whether these states are caused by variations in the host environment or are true alternative states is unclear.

Cycling outcomes are processes where no microbial community is completely resistant to invasion, and the adaptation process continually cycles between a set of states through mutation and invasion events. This cycling can be either periodic (Fig. 3d, *left*) or aperiodic and unpredictable—albeit confined to a closed class of recurrent states (Fig. 3d, *right*). These changes in the community occur at the evolutionary time-scale[35] and, though reminiscent of, should be distinguished from quasi-periodic and chaotic dynamics on ecological time scales[36], which are not driven by mutations. For example, in a dimorphic community of specialists and generalists, mutation and invasion events can drive the generalists to extinction. This extinction leaves an unexploited metabolic niche that a newly-emergent population of generalists can fill, and the cycle repeats (Fig. 3d). Both of these evolutionary outcomes are qualitatively reproduced in stochastic simulations with continuous phenotypes (Supplementary Figs. 3–5).

Visualizing adaptation processes as directed networks not only conveys information about the evolutionary outcomes but also about the dynamics of adaptation. For example, we found that some adaptation processes are bottle-necked: most mutational paths have to transition through a few common states before reaching a recurrent state (Fig. 3e). The network examples we show in Fig. 3 are only representative, and do not portray the entire scope of networks we have observed, but instead illustrate a few properties of adaptation that can be conveyed graphically. We will later quantify such properties of the networks to more comprehensively characterize the dynamics of adaptation and to construct a predictive model.

**Sensitivity of adaptation to environmental conditions**. In light of the multiplicity and complexity of evolutionary outcomes, we investigated how particular outcomes are associated with the environmental parameters of the model. These parameters describe the chemostat's set-up (rate of influx of substrates and dilution rate) and the properties of the two substrates (maximal rate of import, metabolic rate, and yield).

Although some evolutionary outcomes were more likely to be associated with certain types of environments, no clear pattern emerged (Fig. 4a). In particular, we did not detect regions in the space of environmental parameters that robustly associated with a single evolutionary outcome. We first used a nearest-neighbors algorithm[37] to estimate the density of evolutionary outcomes given one environmental parameter with the others varying (i.e., we estimated the probability of each of the eight evolutionary outcomes given the environmental parameter). In general, these relationships were not robust, with the exception of high ratios of the influx rates of the substrates (>4 in log space): no parameter region associated with an evolutionary outcome with probability near 1. Instead, we observed that metabolically diverse (dimorphic) communities are most likely to emerge when the rates of influx for the two substrates are similar and when the dilution rate is low. As the disparity in supply of the substrates increases, we find that communities undergo evolutionary cycling, perhaps signaling the onset of ecological collapse[38]. At even greater ratios of the substrate's rates of influx, communities disappear, leaving a population of a single specialist as the only evolutionary outcome. We also found that equality of the substrate's yields was a necessary condition for the emergence of multi-stationary evolutionary outcomes, which suggests that parity between some properties of the substrates, as for the rates of influx, is important to promote metabolic diversity. However, increasing the difference between either the two metabolic rates or the two maximal rates of import did not lead to notable changes in the probabilities, except for a general trend towards a loss of diversity.

Even by using combinations of environmental parameters to characterize outcomes, substantial unpredictability remained. We developed a hierarchical classification, which combines six statistical estimators to make predictions (Methods section). Its overall performance (a mean recall of 0.78 over the eight evolutionary outcomes), however, did not suggest that a reliable predictive map from environmental parameters to evolutionary outcomes could easily be formed (Supplementary Fig. 6).

**Evolutionary outcomes are interspersed in parameter space**. We can partly understand this unpredictability by considering how the different evolutionary outcomes are distributed in parameter space, where distinct clusters do not appear to exist. For each outcome, we first performed hierarchical clustering using the pairwise Euclidean distance in a standardized parameter space (Methods section) and then aligned these clusters with samples from other outcomes to assess their overlap (Fig. 4b). In general, while parameter sets that produced the same outcomes did form clusters (bright diagonal elements), these clusters were not distinct: a sample or cluster from one outcome is typically proximal to samples or clusters from other outcomes (bright off-diagonal elements). Environmental perturbations are therefore often likely to cause a qualitative shift in the long-term community (Supplementary Fig. 7).

In particular, monomorphic specialists permeate parameter space and are typically close to all other outcomes. By calculating the distribution of shortest distances from one type of outcome to either the same outcome or one of the other seven possibles outcomes, we can estimate the magnitude of the environmental change required to both maintain the type of outcome and to alter one outcome to another (Methods section). Assuming that the probability of environmental perturbation is inversely proportional to its magnitude, a change in the environment is approximately as likely to preserve a diverse community as it is to lead to its collapse to a monomorphic specialist (Fig. 4c, *right*). The converse, however, is not true: a population of monomorphic specialists is far less likely to transition to a diverse community (Fig. 4c, *left*). This asymmetry arises from differences in the cluster shapes in parameter space. As a two-dimensional analogy, consider a circular cluster of outcome $X$ surrounded by a ring of outcome $Y$ of equal area. On average, $Y$ points in the ring will be closer to the $X$–$Y$ boundary than $X$ points in the circle because of the larger interior of the circle. Therefore, small perturbations of $X$ points are less likely than equivalent perturbations of $Y$ points to cause a crossing of the $X$-$Y$ boundary.

We conclude that prediction of evolutionary outcomes from environmental conditions, even in simple chemostats, is likely to be challenging. Evolutionary outcomes are only the endpoints of the adaptive process, however, and we next investigate if the dynamics of these processes (the mutational paths) are characteristic of their endpoints.

**Adaptation dynamics on the network of mutational paths**. The mutational paths of the Markov process describe the dynamics of adaptation, and the processes we construct have tens to millions of mutational paths despite having only eleven discrete phenotypes and a maximum of two co-existing populations in each community.

**Measuring evolutionary repeatability**. We analyze the dynamics of adaptation by enumerating a process's mutational paths and calculating the probability with which each path may be observed to construct the distribution of paths (Methods section). In addition, we calculate several properties of the paths during enumeration (Fig. 5a). For example, we define and calculate the

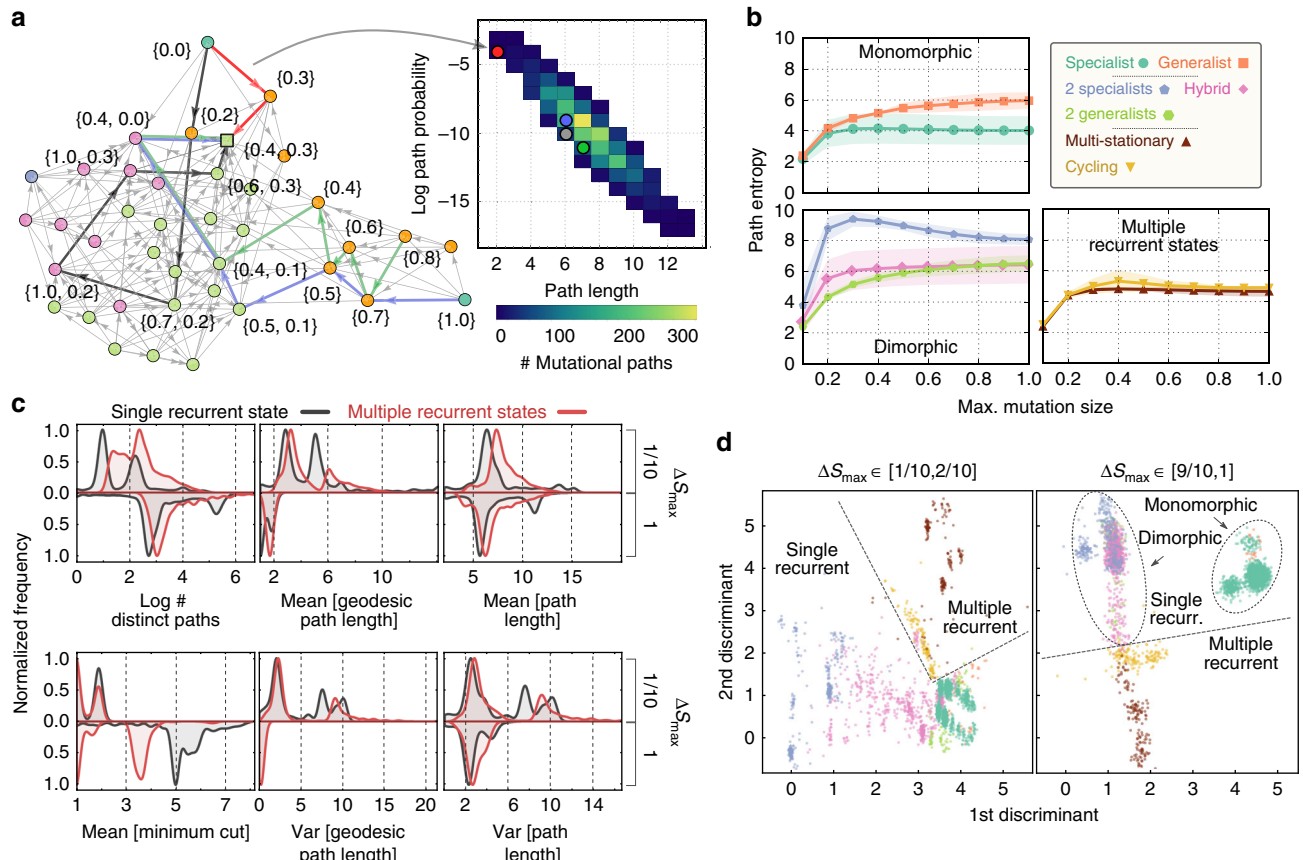

**Fig. 5** Different evolutionary outcomes have characteristic adaptation dynamics. Processes with different evolutionary outcomes have characteristic dynamics of adaptation, which can be quantified and compared through the properties of their mutational paths. **a** We enumerate mutational paths to generate the distribution of paths. Paths begin at a monomorphic state (with equal probability) and end at a recurrent state. A path's probability is inversely proportional to its length. Most mutational paths are of intermediate length, and only a few are either short or long. **b** The repeatability of the dynamics of adaptation depends on the long-term evolutionary outcome and does not typically decrease with larger maximum sizes of mutation. The entropy of the distribution of paths quantifies repeatability, with high entropy implying low repeatability. Points show the mean path entropy; shaded regions are ±s.d. **c** Statistics from the mutational paths vary with both the maximum size of mutation and the evolutionary outcome. We plot the distributions for six properties of the paths, grouped by the number of recurrent states and the two extremes of the maximum size of mutation. The distributions are multi-modal, and the peaks correspond to outcomes lower in the classification of Fig. 3a. **d** Processes with different evolutionary outcomes have characteristic properties of their mutational paths. Through linear discriminant analysis, we identified combinations of the six statistics in **c** that separate the adaptation processes by their evolutionary outcome. A two-dimensional projection of the result is shown with lines and ellipses drawn manually

length of a path as the number of state-to-state transitions in one realization of the adaptation process (a single run of an evolution experiment) from an ancestral to a recurrent state.

The degree to which adaptive evolution is repeatable is of long-standing interest[39, 40], but analyses are typically restricted to models with static fitness landscapes[41]. To quantify the repeatability of adaptation in our model, where fitness landscapes change dynamically through construction and destruction of ecological niches, we mathematically define repeatability as the entropy of the distribution of mutational paths. If replicate experiments are likely to follow only a few mutational paths, the entropy is small and repeatability is high; if each replicate experiment follows a different mutational path, the entropy is large and repeatability is low.

In our model, the repeatability of the dynamics of adaptation is associated with the long-term evolutionary outcome of the adaptation process and varies with the maximum size of mutation (Fig. 5b). With one exception, path entropy does not decrease with the maximum size of mutation as more mutational paths become possible. We find that processes where monomorphic specialists are evolutionarily stable have the most repeatable dynamics, and that this repeatability plateaus early as the

maximum size of mutation increases suggesting that the adapting system can follow only a few mutational paths even as larger mutations become available. In contrast, processes with metabolically diverse communities have the least repeatable dynamics. Notably, dimorphic specialists (and, to a lesser extent, cycling outcomes) have a path entropy that increases to a maximum at intermediate maximum sizes of mutation and decreases thereafter because a few mutational paths with large mutations emerge to dominate the process (Fig. 5b, *blue line*). The rate at which path entropies plateau varies between the different evolutionary outcomes. A non-increasing path entropy suggests that mutational paths with larger mutations are not realized. Frequency-dependent fitness effects may limit large mutations, which lead to phenotypes different from the current resident, because the corresponding mutants are ill-suited to the current environment in the chemostat. Smaller mutations, in contrast, are more likely to gradually alter the environment.

**Mutational path properties identify evolutionary outcomes.** To compare the dynamics of adaptation between processes, we constructed a condensed dynamical profile for each adaptation

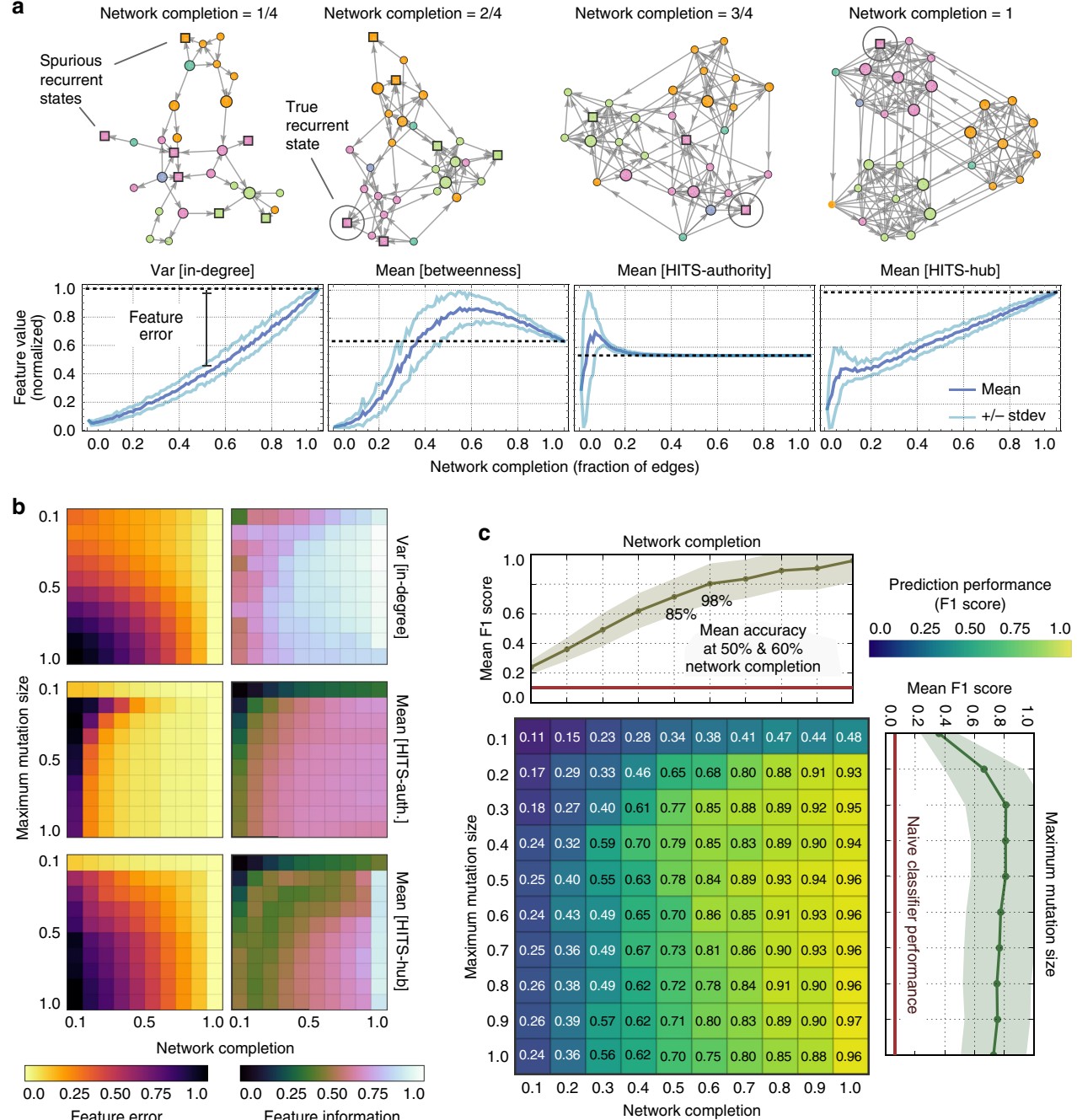

**Fig. 6** Predicting long-term outcomes from the topology of the network of mutational paths. We calculated six centrality measures, which characterize different aspects of the networks' topology, and used these as learning features in a predictive model trained on complete and incomplete networks. **a** The state of a network's completion affects its topology, which is reflected in the statistics of its centralities. As an example, we show a network in various states of completion and the corresponding progression of four statistics (*dark blue* shows the mean and *light blue* shows the s.d. calculated for 100 different networks). **b** We characterized the statistics of the centralities by how they converge to their value when the network is complete (the feature error) and by how well they can discriminate between evolutionary outcomes (mutual information). The plots show the mean of the feature error and the mean of the mutual information between the statistic and the evolutionary outcomes. **c** Evolutionary outcomes predicted with a classifier trained on the statistics of the centralities from partial and complete networks. We assessed performance via the unweighted mean of the *F*1 score taken over the seven outcomes. The mean test-set performance is reported for 10-fold cross-validation. *Top* and *right line* plots show the mean performance, with the s.d.s as *shaded regions*, taken over either the state of completion of the network or the maximum size of mutation. The *red line* is the performance of a naive classifier, which predicts following the empirical frequencies of the outcomes

process through calculating six properties of its mutational paths. These properties were: the number of paths, the mean and variance of the length of the geodesic (shortest) paths, the mean and variance of the length of all paths, and the mean minimum cut size (the smallest number of edges that must be removed from the

graph to disconnect an initial state from a recurrent state and a measure of the extent of bottle-necks in the process).

We analyzed the six properties of the mutational paths for processes with different evolutionary outcomes and at different maximum sizes of mutation (Fig. 5c). The effect of increasing the

maximum size of mutation depends on the evolutionary outcome of the process, but the qualitative trends are consistent. Specifically, larger mutations decrease both the mean and the variance of the length of the geodesic path because recurrent states can be reached via fewer mutation and invasion events of larger effect. Similarly, larger mutations have a decreasing, albeit smaller, effect on the mean and variance of the overall lengths of the paths. Finally, we found that a larger maximum size of mutation generally leads to processes with fewer bottle-necks (higher minimum cut size), particularly for processes with a single recurrent state. Increasing the maximum size of mutation by increasing the availability of mutations can increase the number of outgoing edges from vertices in the network. Mutational paths that pass through such these vertices can often still reach the recurrent states even as edges are removed from the network, and so the degree of bottle-necking is reduced.

The distributions of properties of the mutational paths are multi-modal (Fig. 5c), and the peaks correspond to evolutionary outcomes lower in the hierarchy of evolutionary outcomes (Fig. 3a), suggesting that processes with different outcomes have characteristic dynamics of adaptation. We used discriminant analysis to find linear combinations of the six properties of the paths that maximize the separation of the viable evolutionary outcomes according to their dynamics. We found that different outcomes, particularly those higher in the hierarchy of outcomes (Fig. 3a), occupy different regions in a two-dimensional projection of the discriminant space (Fig. 5d). We can therefore conclude that mutational paths characteristically identify long-term outcomes, and we contrast this positive finding with the difficulty of obtaining an association between environmental parameters and evolutionary outcomes (Fig. 4).

The properties of mutational paths that we calculate quantify statistical aspects of the adaptation process, but enumeration of paths becomes computationally prohibitive for larger networks and relies on the correct classification of states, which itself requires knowledge of the complete network of paths. We next present an alternative graph-theoretic approach that relaxes these constraints and forms the basis of a model for predicting evolutionary outcomes from observations of the dynamics of adaptation.

**Predicting outcomes from networks of mutational paths**. To motivate our predictive model, we consider an evolution experiment as the progressive construction of a network of mutational paths. In such a setting, the resident microbial populations in a chemostat are periodically assayed[42], and the transitions between resident communities are used to progressively construct the network. To simulate the results of this procedure, we re-sampled the 100,000 adaptation processes in our data set to obtain networks at varying stages of completion. Incomplete networks may contain spurious recurrent states, which obfuscate the process's evolutionary outcome (Fig. 6a, top). These states appear recurrent because the mutation and invasion events that lead away from these states have not yet been observed in the experiment. The objective, then, is to forecast the true recurrent states of the complete network from an incomplete sample. While a single adaptive trajectory is unlikely to contain enough information for reliable prediction, we show how incomplete trajectories from replicate evolution experiments can be combined and used in a predictive manner.

**Informative and invariant topological properties**. Our predictive model relies on quantifying how networks change with accumulating observations. As we have shown earlier, evolutionary outcomes have characteristic mutational paths. To use the information encoded in the entire network of mutational paths at once, we measure graph-theoretic properties that quantify abstract features of the network. Specifically, for each network, we calculated six measures of centrality[43] that characterize different aspects of the topology of the network: the in-degree, out-degree, closeness, betweenness, HITS-hub, and HITS-authority centralities (Methods section, Supplementary Fig. 8, and Supplementary Table 1). Though less interpretable from an ecological or evolutionary standpoint compared to the properties of the mutational paths (such as the length of the path), centralities nevertheless contain valuable predictive information. For each network we calculate the mean and variance of the six centralities (taken over the network's vertices). As mutation and invasion events are observed, new vertices and edges are added to the network, and the resulting changes in the network's topology are reflected in changes in the statistics of the centralities (Fig. 6a, bottom).

To quantify how informative is a statistic for a centrality, and therefore how useful the statistic will be in a predictive model, we calculated the mutual information between each of the two statistics for the six centralities and the evolutionary outcomes (Methods section). A high mutual information suggests that processes with different evolutionary outcomes have sufficiently different distributions of centralities to allow us to unambiguously associate a particular statistic of a centrality with an evolutionary outcome (Fig. 6b, feature information). By calculating the difference in the statistics of the centralities between the network at varying stages of completion and when the network is fully complete, we found that some statistics will converge quickly: in the sense that, after a small fraction of the network is discovered, they will remain invariant as more vertices and edges are added (Methods section). In other words, these centralities are abstract topological features of the network whose statistics can be reliably identified with only a few observations (Fig. 6b, feature error). Interestingly, we found that the most informative statistics of the centralities (those with the highest mutual information) were also the slowest in terms of converging to their value when the network is complete (Supplementary Fig. 9).

**Topology predicts outcomes from incomplete observations**. Differences between the topologies of the networks, even for incomplete networks, were sufficient to enable reliable prediction of evolutionary outcomes using a statistical-learning approach. The twelve statistics (the mean and variance of the six centralities) provide a projection of the network of mutational paths, through its topology, to a low-dimensional space. We trained a classifier to predict evolutionary outcomes from these features of the centralities using data from incomplete and complete networks, the degree of the network's completion, and the maximum size of mutation (Methods section). To avoid biasing against rare outcomes, we used the unweighted mean $F1$ score (over the outcomes) as the optimization metric. Neither the degree of the network's completion or the maximum size of mutation will typically be known during an experiment, and we marginalized over both these variables during testing (Supplementary Fig. 10). The performance of the classifier improves with increasing completion of the network and is best at intermediate sizes of maximum mutation (Fig. 6c). At 50% completion of the network, the classifier predicts the correct long-term outcome approximately 85% of the time (average $F1$ score $>0.7$), and at 60% completion predicts correctly 98% of the time ($F1 = 0.8$).

We have shown here that our predictive approach can be useful in forecasting changes in the community through incomplete observations of the dynamics of adaptation, including the loss of (metabolic) diversity. Detecting the onset of transitions in a

community is an enduring problem, but most approaches focus on catastrophic transitions[38, 44]. Nevertheless, the loss of biodiversity following environmental disturbance can, as we have shown, involve a series of ecological transitions mediated by multiple mutation and invasion events, and our network-based approach can address this challenge.

With increasing complexity of ecological models, the computational burden of simulating an exponentially increasing number of combinations of parameters imposes a limit on our understanding of complex and realistic models. When the goal is to forecast long-term outcomes, however, our predictions suggest that it is possible to safely forego simulating the entire model once a sufficient training sample has been obtained. Instead, only a fraction of the mutational path network is necessary to reliably predict the long-term community outcome.

## Discussion

We have shown that a simple model of microbial metabolic adaptation, comprising a trade-off in the use of one substrate over another, can generate surprisingly rich eco-evolutionary behavior. In part this complexity arises because the capacity of an organism to grow depends on the frequency and type of co-existing organisms. In our model, this dependency arises through the creation and destruction of metabolic niches by the differential depletion of substrates by different phenotypes, which has been observed in both experimental evolution[8, 18] and the gastrointestinal tract[45]. Pairwise analysis of microbial interactions may then be insufficient to understand complex communities, and even misleading, because a phenotype's ability to invade depends on the presence of co-residents.

Many of the evolutionary outcomes we see have real-word analogs. For example, the 'mixed' strategy of a single metabolic generalist is consistent with results for yeasts growing on two sugars[10]. Although outcomes of single metabolic generalists were rare in our model, metabolic generalists were more common than specialists in dimorphic communities, being stabilized by the presence of a second population of either generalists or specialists through environmental modification[18]. The multi-stable evolutionary outcomes that we observe have been investigated in multiple ecosystems[46], and alternative microbial communities in the gastrointestinal tract are linked to disease[47]. Although microbial populations that cycle on ecological timescales have been reproduced in the laboratory[36], our cycling evolutionary outcomes are driven by mutations rather than ecological interactions, similar to evolutionary chaos[48].

Experiments in the laboratory reveal that replicate evolving populations have both parallel and unique adaptive changes[17, 18], and we address the repeatability of evolution by calculating the entropy of the distribution of mutational paths. In our model, the probability that two experimental replicates will result in the same sequence of mutations partly depends on the long-term evolutionary outcome. We therefore predict that the repeatability of adaptation can be controlled by choosing environmental conditions. For example, a chemostat with a high ratio for the influx rates for the two substrates is more likely to generate monomorphic specialists, which have the lowest path entropy, and therefore will have lower variability between replicates than a chemostat with a low ratio of the influx rates.

The behaviors we observe are sufficiently complex to preclude reliable prediction of evolutionary outcomes from environmental conditions, even qualitatively. Such prediction is therefore likely to be sensitive to the accuracy of experimental measurements. A further corollary is that an environmental perturbation is more likely to lead to the collapse of a microbial community than the emergence of a more diverse community. Unlike catastrophic

transitions[38, 44], this loss of biodiversity can involve multiple mutation-and-invasion events, which may mask the long-term impact of an environmental change.

Nevertheless, we demonstrate that both statistical properties of mutational paths and topological properties of networks of these paths allow the characterization of evolutionary outcomes. We therefore highlight the dynamics of adaptation as a key variable for prediction and emphasize that this association could also work in reverse: it may be possible to infer the properties of mutational paths by observing the outcome of an adaptive process. Our predictive model relies on the topological properties of the networks of mutational paths, which we summarize through the statistics of centralities of the networks. These statistics therefore have a predictive utility similar to structural indicators in evolutionary game theory[49, 50] and statistical indicators in ecological early-warning systems[38, 44].

The analysis of networks is increasingly being adopted in ecology and evolution. In microbial ecology, data for the abundance of species is used to reconstruct the network of microbial interactions in a community[51]; in evolutionary game theory, the effect of a population's structure on the spread of mutations is studied on networks of connected individuals[52]. Our approach is complementary but different, connecting states of communities rather than of species or individuals, and facilitates the visualization of adaptation on dynamic fitness landscapes, as well as the application of techniques from graph theory and machine learning to the analysis and prediction of adaptation.

Our modeling approach combines elements from the replicator-mutator equation[53, 54] and adaptive dynamics[11, 12, 32] to couple ecological dynamics to an evolutionarily process. Unlike replicator-mutator models, we consider only rare mutations, where a less fit phenotype cannot be maintained through a constant contribution of mutating individuals from a more fit phenotype[55]. We follow adaptive dynamics in treating ecological invasion after mutation as the fundamental unit of adaptation. Nevertheless, mutations in our model are not infinitesimally small and adaptive trajectories need not be continuous in phenotype space. We show that the theory of stochastic (Markov) processes is a suitable framework for analyzing both the adaptation dynamics and long-term evolutionary outcomes, and the invasion maps we produce can be thought of as higher-dimensional analogs of the standard pairwise invasibility plots[12].

Although we capture many eco-evolutionary phenomena, our models simplify several aspects of the evolution and function of communities. The separation of ecological and evolutionary timescales should be relaxed to allow multiple, simultaneously invading mutants[56] and mutant populations with multiple co-occurring mutations[57, 58]. Both effects are often pervasive in microbial evolution[59]. We only consider exploitative competition between members of a microbial community, but other interactions are possible, such as cross-feeding and commensalism[51]. Potential phenotypic changes are constrained by the types of genetic mutations that are possible[60], and we have modeled such constraints coarsely. More sophisticated models, going beyond the uniform distribution within a maximum size of mutation that we consider, should change both the properties of the mutational paths and their networks and, potentially, the nature of the possible long-term evolutionary outcomes. Finally, our phenomenological treatment of metabolic specialization could be replaced with a data-informed genotype-phenotype map; for example, flux balance analysis can predict a microbe's metabolic phenotype from its metabolic genotype[61].

Advances in laboratory evolution make microbes well-suited for validating our predictions. A library of mutants for either a particular gene or trait of interest could be generated through random mutagenesis[62]. To determine the outcome of all possible

invasions, perhaps following a dynamic algorithm to direct and reduce the number of necessary experiments (Fig. 2), the mutants could be co-cultured and the results combined to construct empirical networks of steady-state microbial communities. Such networks could be directly compared to those generated *in silico*.

In conclusion, we present a model that demonstrates the importance of trade-offs for generating metabolic complexity in microbial communities[1, 5]. If a cell is able to evolve its response to one substrate unconstrained by its response to the other, then only one evolutionary outcome is possible: a phenotype that maximally depletes both substrates. Metabolic trade-offs, however, preclude this single optimal phenotype and together with dynamic environmental niches[2, 8, 18] and a limited distribution of (possibly large effect) mutations[63] generate intricate dynamics of adaptation and long-term behaviors. Intracellular trade-offs are expected to be common[26, 64, 65], and our results support the idea that the resulting frustration of optimal responses is a major factor generating the complexity observed in microbial communities.

## Methods

**Modeling the chemostat with two substitutable substrates**. The model of the chemostat is the ecological component of our eco-evolutionary framework. Assuming rare mutations, we consider the model's ecology in the absence of evolutionary change and only later incorporate an evolutionary process. The environment in the well-mixed chemostat is spatially homogeneous, and the abundance of the biotic components is modeled over time using ordinary differential equations[21]. We include a second substrate that is perfectly substitutable with regards to the first[5, 22]: either substrate is sufficient for growth. We name the substrates $u$ and $v$.

To include growth, we structure microbial populations into $N_s$ states of growth that a cell must pass through to replicate. A cell can exist in any one of the possible growth states and in all of these states can bind either a $u$ or a $v$ substrate, but not both simultaneously. Let $\mathbf{n}_x(t) = (n_{1,x}(t), n_{2,x}(t), \ldots, n_{N_s,x}(t))$ be the abundance of cells with phenotype $x$ in each growth state and $\mathbf{n}_x^{(u)}(t)$ and $\mathbf{n}_x^{(v)}(t)$ be the corresponding vectors for when a cell is bound by and is metabolizing $u$ and $v$. The system of ordinary differential equations describing the ecological dynamics can then be written in matrix notation as

$$
\begin{aligned}
\dot{\mathbf{n}}_x(t) &= -\mathbf{n}_x[uk_u s_x + vk_v(1-s_x) + D] \\
&\quad + m_u[\mathbf{\Gamma}^{(u)}]^T \mathbf{n}_x^{(u)} + m_v[\mathbf{\Gamma}^{(v)}]^T \mathbf{n}_x^{(v)} \\
\dot{\mathbf{n}}_x^{(u)}(t) &= \mathbf{n}_x uk_u s_x - \mathbf{n}_x^{(u)}(m_u + D) \\
\dot{\mathbf{n}}_x^{(v)}(t) &= \mathbf{n}_x vk_v(1-s_x) - \mathbf{n}_x^{(v)}(m_v + D) \\
\dot{u}(t) &= u^{(0)} - u\left[D + k_u \sum_x \mathbf{1}^T \mathbf{n}_x s_x\right] \\
\dot{v}(t) &= v^{(0)} - v\left[D + k_v \sum_x \mathbf{1}^T \mathbf{n}_x (1-s_x)\right]
\end{aligned}
\tag{1}
$$

where: $k_u$ and $k_v$ are the maximal rates of import for $u$ and $v$; $m_u$ and $m_v$ are the substrate-specific metabolic rates; $u^{(0)}$ and $v^{(0)}$ are the influx concentrations for the two substrates; and $D$ is the chemostat's dilution rate. The metabolic specialization of phenotype $x$ is parametrized by $s_x$, a value between 0 and 1.

Progression through the growth states is encoded in the $\mathbf{\Gamma}^{(u)}$ and $\mathbf{\Gamma}^{(v)}$ matrices and depends on the yields of each substrate, $\gamma_u$ and $\gamma_v$, which are measured in increments to the growth state. Transitions from a state $p$ to a state $q$ are proportional to:

$$
\Gamma_{p,q}^{(u)} = \delta_{p+\gamma_u,q} + \delta_{p+\gamma_u-N_s,q} + \delta_{p,1}\theta(p + \gamma_u - N_s)
$$

$$
\Gamma_{p,q}^{(v)} = \delta_{p+\gamma_v,q} + \delta_{p+\gamma_v-N_s,q} + \delta_{p,1}\theta(p + \gamma_v - N_s)
$$

$$
\tag{2}
$$

where $\delta_{p,q}$ is the Kronecker delta and $\theta(z)$ is the Heaviside function. In both equations, the first term describes a cell transitioning directly from state $p$ to state $q$ through metabolizing a $u$ or a $v$ substrate. The second and third term are non-zero when a cell metabolizes enough substrate to divide (increment its growth state past $N_s$): the second term is the leftover 'mass' in the mother cell that determines its new growth state following replication, and the third term describes the birth of a new daughter cell (and therefore can only be non-zero for $q = 1$).

We note that if the size (the number of phenotypic values) of the space of discrete phenotypes is $N_p$, then the model has $N_p$ equations for each of $\mathbf{n}_x$, $\mathbf{n}_x^{(u)}$, and $\mathbf{n}_x^{(v)}$, following Eq. (1). We have chosen $N_p = 11$ throughout.

**Simulating invasion by a mutant phenotype**. To resolve the outcome of invasion and competition between any combination of residents and a mutant, we numerically integrated the model (Eq. (1)) for all phenotypes to steady-state after including a small mutant population.

Let $N_x^*$ denote the steady-state abundance of the microbial population with phenotypic value $s_x$, i.e.,

$$
N_x^* = \sum_{i=1}^{N_s} \left[ n_{i,x} + n_{i,x}^{(u)} + n_{i,x}^{(v)} \right].
\tag{3}
$$

We consider vectors over all $N_p$ phenotypic values (over all discrete values of $s_x$), and at most two elements of these vectors are non-zero at steady-state because at most two phenotypes can co-exist at steady-state because we have two substrates. If the steady-state before invasion by a mutant is

$$
\left( N_0^*, N_1^*, \ldots, N_{N_p}^*, u^*, v^* \right),
\tag{4}
$$

then when a nascent mutant with phenotype, $y$, emerges, we perturb this steady-state to include a small population of mutants with abundance $\epsilon$. The magnitude of $\epsilon$ is determined by multiplying the steady-state abundance of the smallest, positive resident by a constant factor, $\delta$. If the chemostat contains no resident, we set $\epsilon$ to $\delta$.

The model, Eq. (1), is numerically integrated to steady-state using the initial condition

$$
\left( N_0^*, \ldots, N_y^* + \epsilon, \ldots, N_{N_p}^*, u^*, v^* \right),
\tag{5}
$$

and this steady-state will be different from the original steady-state if the invasion is successful. Numerically, we define a steady-state to be when the relative magnitude of all derivatives is smaller than a threshold: $\left| \frac{\dot{n}_{i,x}}{n_{i,x}} \right| < 10^{-3}$.

We use the composition of extant phenotypes to distinguish the different steady-states of the ecological model. In our model, like others[66], there is one unique, globally attracting steady-state[67]. For simplicity, we represent the composition of extant phenotypes (indexed by $x$) in a steady-state $k$ by a binary vector of length $N_p$, $\mathbf{C}^{(k)}$, where $C_x^{(k)} = 1$ if $N_x^* > \epsilon$ and $C_x^{(k)} = 0$ otherwise. Here $\epsilon$ has the same magnitude as the $\epsilon$ used for the initial size of the populations of mutants. This choice means that neutral mutations, which neither grow or decline when they first emerge in some resident community, will not be preserved in our chemostat; therefore, our model of adaptive evolution does not include neutral evolution.

**Eco-evolutionary model in the weak-mutation limit**. We calculate solutions to all possible invasions that can arise in the eco-evolutionary setting using dynamic programming and the results are stored in an invasion map (Fig. 2a and Supplementary Note 3).

We next construct the discrete-time, embedded Markov chain in the limit where the ecological and evolutionary timescales are perfectly separated (following adaptive dynamics[11, 12, 32]). Assuming that the process has phenotypic vector $\mathbf{C}^{(i)}$, then the probability of a transition to state $\mathbf{C}^{(j)}$ from a single mutation and invasion event in a small unit of evolutionary time, $\Delta t$, is

$$
p_{j|i}(\Delta t) = \sum_{x \in \mathbf{C}^{(i)}, y \notin \mathbf{C}^{(i)}} N(x|\mathbf{C}^{(i)}) b(x|\mathbf{C}^{(i)}) \mu(x|\mathbf{C}^{(i)}) M_{y|x} I(y|\mathbf{C}^{(j)}, \mathbf{C}^{(i)}) \Delta t
\tag{6}
$$

for $j \neq i$. In Eq. (6), $N(x|\mathbf{C}^{(i)})$ is the abundance of the phenotypes in the population indexed by $x$ at steady-state $i$; $b(x|\mathbf{C}^{(i)})$ is the per capita birth rate of population $x$ at steady-state $i$, which in the chemostat is always equal to the dilution rate; $\mu(x|\mathbf{C}^{(i)}) = \mu$ is the mutation probability per birth, which we assume is constant; $M_{y|x}$ is the probability that a mutant offspring of a cell with phenotype $s_x$ will instead have phenotype $s_y$. We call $\mathbf{M}$ the phenotype adjacency matrix, which is symmetric and uniform and defined by the maximum size of mutation size, $\Delta S_{max} \in (0, 1]$. We first write $\bar{M}_{y|x} = 1$ if $|s_x - s_y| \leq \Delta S_{max}$, and $\bar{M}_{y|x} = 0$ otherwise, and then row-normalize to include boundary effects so that $M_{y|x} = \bar{M}_{y|x} / \sum_z^{N_p} \bar{M}_{z|x}$. Finally, $I(y|\mathbf{C}^{(j)}, \mathbf{C}^{(i)})$ is the probability that a rare $s_y$ phenotype invading a resident of phenotypic composition $\mathbf{C}^{(i)}$ will transform the composition to $\mathbf{C}^{(j)}$ and is either 0 or 1 because our ecological models are deterministic (Eq. (1)). Nevertheless, stochastic extinction because of a small initial number of mutants can be incorporated into this term.

We need only consider transition probabilities at ecological steady-states. When mutations are rare, the chemostat will be at steady-state when a new invasion and mutation event occurs and transition probabilities will evaluate to zero in the limit $\Delta t \to 0$ otherwise. Adaptation in the weak-mutation limit can be modeled as a 'jump' process, where time is interpreted in terms of successful mutation and invasion events, or jumps.

To calculate the transition probabilities, we first form the infinitesimal generator matrix, $Q$, by calculating the transition rates, $q_{j|i}$:

$$q_{j|i} = \lim_{\Delta t \to 0} \frac{p_{j|i}(\Delta t)}{\Delta t} \text{ for } i \neq j \quad \text{with} \quad q_{i,i} = -\sum_{j \neq i} q_{j|i} \qquad (7)$$

From Eq. (7), we can find the transition probabilities, $t_{j|i}$, for the discrete-time (embedded) Markov process. If $q_{i|i} = 0$ then $q_{j|i}$ is zero for all $j$ from Eq. (7), and there is no flow of probability from state $i$. Hence

$$t_{i|i} = 1 \quad \text{and} \quad t_{j|i} = 0 \quad \text{for } q_{i|i} = 0. \qquad (8)$$

If $q_{i|i} \neq 0$, then there is flow of probability away from state $i$ and

$$t_{i|i} = 0 \quad \text{and} \quad t_{j|i} = q_{j|i} / \sum_{k \neq i} q_{k|i} \quad \text{for } q_{i|i} \neq 0. \qquad (9)$$

Finally, we assume that the probability distribution for the initial state is uniform over all monomorphic states providing that those monomorphic states can exist in the chemostat without competition.

**Predicting outcomes from environmental parameters**. We used a hierarchical scheme to classify each process of adaptation by its qualitative evolutionary outcome (Supplementary Fig. 6). Our best attempt at predictive required training a hierarchy of (mostly) binary classifiers. This model follows the hierarchical scheme of evolutionary outcomes in Fig. 3a: for each of the six non-leaf (non-terminal) nodes we optimized, trained, and tested either a support vector machine classifier with a kernel of radial basis functions or a random forest classifier. The best-performing classifier at each node was included in the final model, which makes predictions by percolating a sample down the hierarchy until it encounters a leaf node. For the ternary node (from 'dimorphic state' to 'specialists' or 'hybrid' or 'generalists'), we optimize three one-vs.-one random forest sub-classifiers. Classification in the ternary setting is achieved via majority voting[37].

To optimize node classifiers, we used a random search of hyperparameters. Hyperparameters were randomly sampled from specified distributions, and the classifier was trained and tested on a nested (stratified) partitioning of the training set in a 5-fold manner to obtain a mean score of optimization. The classifier's performance during optimization was calculated from the area under the receiver operating characteristic curve[37]. The estimator with the highest scoring set of hyperparameters was chosen, re-trained on the entire training set, and was subsequently tested against the original test set. This process was repeated 10 times to obtain the mean test score for the node classifier (Supplementary Fig. 6).

**Outcome clusters and shortest-distance distributions**. We calculated the pairwise Euclidean distances between all standardized parameter sets. After log-transforming the rate of influx of the substrates, the maximal rates of import, the metabolic rates, and the chemostat's dilution rate, we standardized the environmental parameters to have zero mean and unit variance to normalize for differences in scale.

To discover within-outcome groupings, we hierarchically clustered samples with the same evolutionary outcome using the Euclidean distance and Ward's method for joining clusters. To assess the degree of overlap between clusters with different outcomes, we used the within-class indexing returned by the clustering algorithm to re-sort the rows and columns of the pairwise distance matrix. Not only does this re-sorting move together samples with the same evolutionary outcomes that are close in parameter space (as bright boxes along the diagonal in Fig. 4b), but also reveals how dissimilar these clusters are with respect to clusters of other evolutionary outcomes (off-diagonal blocks Fig. 4b).

To calculate the distribution of shortest distances between evolutionary outcomes (Fig. 4c), we standardized the parameter space as before and used the Euclidean distance to determine each sample's nearest neighbor for each of the eight evolutionary outcomes.

**Mutational path distribution and path properties**. We search a network to find all paths, $p$, between an (ancestral) state and the set of recurrent states for that network. To avoid infinite paths in the case of cycling processes, we terminate the relevant branch of the algorithm when any recurrent state is encountered. We keep a list of probabilities for the state-to-state transitions in each path. The number of transitions in a path, $p$, is its length, $L_p$, and the probability of the path is

$$h_p = P_{\text{ancestral}}(i_0) \left[ t_{i_1|i_0} t_{i_2|i_1} \dots t_{i_{L_p}|i_{L_p-1}} \right] \qquad (10)$$

where $P_{\text{ancestral}}(i_0)$ is the probability that adaptation starts at state $C^{(i_0)}$ and $t_{i_1|i_0}$ is the transition probability from state $C^{(i_0)}$ to state $C^{(i_1)}$, and so on. Note that the $h_p$ define a probability mass function, i.e., $\sum_{p \in \mathcal{P}} h_p = 1$, where $\mathcal{P}$ is the set of all unique paths, because the process must start at one of the initial ancestral states and be absorbed in a recurrent state. We define the geodesic paths, $\mathcal{G}$, as those paths, $g$, with initial state $C^{(g')}$, and final (recurrent) state $C^{(g'')}$ whose length is the minimum possible for a path between $C^{(g')}$ and $C^{(g'')}$.

Summary statistics (mean and variance) for the properties of the paths are calculated using either the distributions of paths ($h_p$) or of geodesic paths ($h_g$), where we normalize to find the probabilities conditioned on the geodesic set: i.e., $h_g = h_g / \left( \sum_{\tilde{g} \in \mathcal{G}} h_{\tilde{g}} \right)$.

**Centrality convergence (error) for incomplete networks**. To characterize how a statistic of a centrality converges to its value when the network is complete as a function of the degree of a network's completion, we used a scale-free measure of error. Let $x(c, \Delta S_{\text{max}})$ be the value of the statistic for networks at completion $c$ (fraction of edges of the complete network included in the incomplete network) and maximum size of mutation $\Delta S_{\text{max}}$. Then let

$$y(c, \Delta S_{\text{max}}) = \frac{x(c, \Delta S_{\text{max}}) - \min_{c,\text{samples}} \{x(c, \Delta S_{\text{max}})\}}{\max_{c,\text{samples}} \{x(c, \Delta S_{\text{max}})\} - \min_{c,\text{samples}} \{x(c, \Delta S_{\text{max}})\}} \qquad (11)$$

be the normalized value of $x(c, \Delta S_{\text{max}})$. The minimum and maximum are taken over all values of completion, $c$, and over all samples and so establish the observed range of values of the statistic. This normalization ensures that $y(c, \Delta S_{\text{max}}) \in [0, 1]$ for all samples. We defined the normalized error for this sample process as

$$e(c, \Delta S_{\text{max}}) = |y(c, \Delta S_{\text{max}}) - y(1, \Delta S_{\text{max}})| \qquad (12)$$

where $y(1, \Delta S_{\text{max}})$ is the normalized value of the statistic when the entire graph is discovered (i.e., when graph completion is 1 and all edges are known). We define the feature convergence error as the mean of $e(c, \Delta S_{\text{max}})$ taken over all samples in the data set of partial graphs. We repeat this procedure for all twelve centrality statistics (Supplementary Fig. 9).

**Mutual information between centralities and outcomes**. To characterize how well a feature discriminates between processes with different ultimate evolutionary outcomes, we estimated the mutual information between the feature and the ultimate evolutionary outcome. We are interested in this ultimate evolutionary outcome because the evolutionary outcome of a process's incomplete networks may be different from the true outcome when the network is complete. We grouped statistics by the degree of completion of the networks and by maximum size of mutation to assess whether some centralities are more or less informative. Accordingly, the mutual information, $I(X, Y; c, \Delta S_{\text{max}})$, between each of the twelve statistics, $X$, and the ultimate evolutionary outcome, $Y$, was calculated as

$$I(X, Y; c, \Delta S_{\text{max}}) = H(X(c, \Delta S_{\text{max}})) - (HX(c, \Delta S_{\text{max}})|Y(c, \Delta S_{\text{max}})) \qquad (13)$$

where $X(c, \Delta S_{\text{max}})$ is the distribution of the statistic of the centrality and $Y(c, \Delta S_{\text{max}})$ is the distribution of the ultimate evolutionary outcomes, constructed from samples of complete and incomplete networks that have been grouped by the degree of completion of the network ($c$) and the maximum size of mutation ($\Delta S_{\text{max}}$). Note that, because we work with the ultimate evolutionary outcome of an incomplete network: $Y(c, \Delta S_{\text{max}}) = Y(1, \Delta S_{\text{max}})$ for all $c$. We used a nearest-neighbors procedure to estimate the density[68] to calculate both $H(X(c, \Delta S_{\text{max}}))$ and $H(X(c, \Delta S_{\text{max}})|Y(c, \Delta S_{\text{max}}))$. We reported the normalized variant of mutual information, which has a maximum value of 1 when the evolutionary outcome can be perfectly predicted from the value of the centrality's statistic: $I(X, Y; c, \Delta S_{\text{max}})/H(Y(c, \Delta S_{\text{max}}))$.

**Predicting evolutionary outcomes from incomplete networks**. To predict evolutionary outcomes from samples of incomplete networks, we trained a machine-learning model on the networks' topological properties. Centralities characterize aspects of networks' topologies[43], and we focused on six well-known centralities: in-degree, out-degree, closeness, betweenness, HITS-hub, and HITS-authority (Supplementary Fig. 8). Centralities are calculated on a per-vertex basis, and we summarized the distribution of each centrality for each network by its mean and variance.

We used the resulting twelve statistics, as well as the maximum size of mutation and the degree of completion of the network, as features in a random forest classifier. The model was evaluated in 10-fold cross-validation using the unweighted $F1$ score mean (over the evolutionary outcomes) as the performance metric to reduce bias against rare evolutionary outcomes. We used a random search to optimize the hyperparameters of the classifier in a nested manner. The training and testing procedure is summarized below:

1. Partition the entire data set of complete and incomplete networks into 10 stratified folds, where each fold contains roughly similar fractions of each class of outcomes. In addition, we partitioned the data such that the training and testing networks did not overlap, even at different degrees of completion of the networks: we did not train on a complete network and then test the model on an incomplete sample of the same network.
2. For each of the 10 outer folds (above), we further partition the training data to 10 stratified inner folds, following the same rules as before.
3. We then randomly select a set of hyperparameters for the classifier and evaluate its performance by training and testing on the inner folds. We randomly choose $10^6$ incomplete networks for training each time because

even the innermost training set was too large to fit in memory (~$5.7 \times 10^7$ networks). We repeat this nested cross-validation method to identify suitable hyperparameters.

4. The optimal set of hyperparameters (from inner cross-validation) is then used to train the classifier on the training data from the outer fold.
5. The trained model is then tested on the test data from the outer fold.
6. Steps 2–5 are repeated on the next outer fold.

To closely assess the performance of the classifier, we report the performance on the outermost test set averaged over the 10 outermost folds and grouped by the degree of completion of the networks and the maximum size of mutation.

Although training and optimization was done using the 12 statistics, the degree of completion of the networks, and the maximum size of mutation, we reasoned that the degree of completion of the networks and the maximum size of mutation will usually be unknown. To predict evolutionary outcomes without knowing these two quantities, we repeated the above procedure, but now marginalized over both quantities using a uniform prior and using the outcome with the highest marginal probability as the classifier's output (Supplementary Fig. 10).

**Code availability**. The computer code used to generate and analyze the data in this study is available from the corresponding author upon request.

**Data availability**. The data that support the findings of this study are available from the corresponding author upon request.

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

## Acknowledgements

We thank Elco Bakker, Matteo Cavaliere, and especially Luke McNally for critical comments on the manuscript and Nikola Popovic for assistance with mathematical analysis. We acknowledge support from a Wellcome Trust PhD studentship (C.J.), the Scottish Universities Life Sciences Alliance (P.S.S.), and the Human Frontier Science Program (C.J. & P.S.S.).

## Author contributions

C.J. and P.S.S. conceived research. C.J. performed research and analyzed data. C.J. and P.S.S. wrote the paper.

## Additional information

**Competing interests:** The authors declare no competing financial interests.

