## [Peer Review File · Nature Communications]

Reviewers' comments:

Reviewer #1 (Remarks to the Author):

Josephides and Swain generate a novel computational approach for investigating how tradeoffs influence adaptive dynamics. Specifically the authors use a Markov process to generate maps of all possible evolutionary trajectories under different environmental conditions. The approach differs from previous approaches by allowing larger mutational steps and maintenance of two genotypes. The primary findings are that 1) there are many possible evolutionary outcomes even in simple environments 2) evolutionary outcomes cannot be predicted from environmental conditions and 3) limiting mutation size tends to make evolutionary dynamics more repeatable, but increases the number of steps to an evolutionarily stable endpoint.

The authors provide a large amount of data in this densely packed manuscript. The richness of evolutionary dynamics is interesting. My primary concern is that the authors do so little to connect their work to the field more broadly. For example, how do any of the computational results compare to wet lab observations? Further, it is important to highlight that the new method recapitulates previous results (i.e. Supp figures 4 and 5 should be discussed more). A vast array of network analysis is carried out, but this analysis is weakly connected to biological understanding.

Minor suggestions:

- 1) The data in figure 5C is difficult to see. Perhaps darker colors could be used?
- 2) I found the direction of the y-axis in figure 6B confusing, but this is a personal preference.
- 3) It would be nice to see some discussion of how mutational constraints might influence changes in phenotype over evolutionary time. I am not suggesting more analyses, just discussion of the fact that the phenotypic changes that are being modeled would actually be driven by genetic mutations. Some mutations may be more common than others, and some phenotypes may be mutationally inaccessible (reversions may be particularly unlikely). Additionally, evolvability/robustness of organisms might change as a result of adaptation.
- 4) In the last section of the results it would be useful to see more discussion of how incomplete networks relate to a single adaptive trajectory. A single adaptive trajectory seems different than partial mapping of the complete network of all possible evolutionary trajectories. I suspect that one would get much less information from a single adaptive trajectory than is being considered in the partial networks.

Reviewer #2 (Remarks to the Author):

In the manuscript, authors Josephides and Swain propose a simple model on the competition and the accompanying adaptive evolution of microbial communities, and perform *in silico* experiments on it to shed light on its dynamics and emerging patterns. They hierarchically separate the evolutionary outcomes of this model into seven broad categories defined in terms of the phenotypes of the constituent populations, and study how these outcomes depend on a) environmental parameters, b) the structure of the network of mutational paths that represent the dynamics of the adaptation process. Taking one further step in the latter direction, they make the argument that in more complex models that involve more environmental parameters and more diverse populations, the computational burden might prevent the comprehensive simulation of the entire system (i.e. the determination of all possible mutational paths) and therefore predicting those outcomes on incomplete networks becomes of essence. They demonstrate that this is possible using simple centrality measures.

In general, I found the paper well-written and the subject matter compelling. The suggested model is simplistic but the resulting dynamics are rich, which lends itself well to a system-wide treatment by the use of graph theoretical tools. Networks are increasingly being used to represent associations between microbial communities, and the extension of the same approach to networks representing temporal steps of evolution presents an exciting direction for researchers. That said, the paper has considerable room for improvement to make it more accessible to the general audience. Below are my specific and general comments.

1) As a general comment, it may be a little hard at times to follow the text because some eco-evolutionary concepts and terms are used without much introduction. As it is now, it looks more like it is aimed at a topic-specific journal than the broad readership of *Nature Communications*. It could benefit from having some basic concepts, even if they're completely fundamental to readers in the field, defined in one sentence in order to prevent it from sounding like jargon. For example, there could be a brief definition of chemostat. It's simple enough a concept - just needs to be described shortly. The same goes for "import capacity" on line 68. Along the same lines, the Fig. 1C paragraph (lines 82-86) can be made much more "user-friendly". The use of terminology like "substrate specific yield" (instead of mass, I guess?), etc. makes it hard to follow whereas the basic premise is quite simple. Cells eat u and v at rates proportional to u and v concentrations and divide once a certain mass is reached. I think all of the above can simply be circumvented by defining "import rate", "metabolic rate", "yield", "growth state" etc. in simple terms once and for all, in Figure 1's caption.

2) The waiting time between state transitions is ignored in the paper. Can the authors comment on whether or not the time elapsed between transitions has any implications in terms of the evolutionary time scale?

3) Since it's a frequently used parameter in the subsequent analyses, it would be helpful to the reader to include a short biological interpretation of the maximum mutation size Δ_S with a few words, alongside its formal definition in line 125.

4) Could the authors make the explanation of truncation a little clearer? What exactly is meant by recursion?

5) In line 146, what exactly are environmental parameter sets? I understand that it's 10,000 realizations with different parameters but it would be helpful to reiterate which environmental parameters were changed.

6) In Fig. 3, four out of 10,000 environmental parameter sets are shown. How representative are these? The authors show the statistics of these to some extent in the following sections but how can one make sure that the four types are the only patterns of interest? Is there a way to quantify that?

7) In lines 181-182, the authors say no clear pattern emerged in Fig. 4A. What was the hypothesis, what kind of an outcome was expected? In other words, how should each of these plots be interpreted to see any "pattern"? The first two panels are (dilution rate and influx) are discussed but what about the interpretation of the last three panels? Also, how are the dotted lines in the second panel of Fig. 4A determined? Are they a guide to the eye or quantified?

8) Supplementary Fig. 9: How exactly is it decided that it's poor predictive performance (not reliable) with the given recall rates? They look high to me. How is the mean 0.78 when the lowest recall is 0.83? It may be something obvious that I cannot see, but what was the criterion here? It would help if this were clarified.

9) It's important to note that Supp. Fig. 10 is not symmetric, i.e. distinguishes between the "from" state and the "to" state. The same applies to Fig. 4C. In fact, both figures can be made easier to understand, especially Fig. 4C, which is a main figure. It's confusing to have the labels in the same order in the left and right panels. It should simply say distance from monomorphic specialist to other in the left, and other to monomorphic in the right.

10) On a related note, it's not entirely clear to me why Fig. 4C is not symmetric if it's based on distance. Can the authors clarify the rationale behind this asymmetry? It makes sense that it's much more likely to transition from a diverse population to a specialized one than the other way around, but how does that translate to what we see in the Euclidean distance distributions? A simple interpretation would help the reader a lot here.

11) From Fig. 5B, it looks to me as if it's not only dimorphic specialists but also multiple recurrent and monomorphic specialist that don't increase monotonically but rather plateau early on and stay the same with increasing mutation size. The authors, however, say that entropy typically increases with mutation size. That seems to apply only to orange (one generalist), green (two generalists) and hybrid (to some extent).

12) In my opinion, Fig. 5A does a better job at simply explaining how to read the mutational path networks (i.e. starts from a monomorphic state, ends at a recurrent state etc.). This explanation should be carried over to Fig. 3B-E as well.

13) Line 247: Larger mutation size leads to fewer bottlenecks. Could the authors shortly explain what may be reason like they did in the same paragraph for the path lengths?

14) Supplementary Fig. 11 needs a much more informative caption. The mean and variance of each measure for each network is given but overall the figure doesn't convey to the reader how these centrality measures differ qualitatively from each other. It says, "vertices are scaled..." to mean the vertex size, this should be clarified. At minimum, it should be made clear that, simply, the larger nodes, be it round or square, have higher centrality of the respective kind. Although an even better approach would be to give specific network instances that emphasize each centrality type, e.g. a bottleneck node for high betweenness centrality.

15) Line 287: "Most informative centrality measures converged gradually?" What is meant by most informative, simply ones with the highest mutual information? In any case, the authors should try to explain intuitively why in-degree of HITS hub are considered more informative, and what it means for them to converge gradually? Does it mean they are more crucial as a centrality measure to predict true recurrent states? If so, this should be stated explicitly to help the reader understand this section better.

16) As a final suggestion for a discussion point, I think the authors could recognize some recent efforts on network methods used in microbial evolution modeling such as: Faust, K., & Raes, J. (2012). Microbial interactions: from networks to models. *Nature Reviews Microbiology*, 10(8), 538-550.

I think it's important to comment on the current divide between simple computational models and real-world microbial communities and how current methods can be improved specifically in this direction to align the two worlds.

Minor issues and typos:

1) Typo in Figure 2 caption, line 3: "phenotypes values"

2) In Supplementary Figs. 1 and 2, m_u and m_v are not defined.

3) What's the circle next to "truncated node" in Fig. 2? I believe it means darker gray nodes are truncated nodes, this should either be made clear or it should be removed.

4) Line 153: A (insert: single) recurrent state may be a dimorphic or...

5) Fig. 4B says both "normalized Euclidean distance" on the color bar and "pairwise standardized parameter distance" on the right hand side. Are they not the same? It would be better to standardize the notation.

6) Line 182, "Using a nearest-neighbors algorithm..." The authors cite the original paper for this algorithm in the methods section so there should be a reference here to the Methods

section.

7) Supplementary Fig. 10: I am assuming Euclidean distances, i.e. similarities, are meant by "normalized shortest distances". If so, this should be made clear so as not to confuse the reader since the term "shortest distance" has a clear connotation in network theory. The shortest distances mentioned here could be confused with the shortest paths or geodesics in the network.

8) Figure 2 caption, line 3: "phenotypes values"

Response to reviewers
Predicting metabolic adaptation
from networks of mutational paths

Christos Josephides, Peter S. Swain

We would like to thank both reviewers for their critical comments, which we believe have substantially improved the paper. We have re-written the Discussion section and substantially revised the Methods and Supporting Information sections; elsewhere in the manuscript, we have marked significant changes in blue. The reviewers' reports are reproduced in bold below wherein we interpose our response to individual points as they appear.

Response to reviewer 1

Josephides and Swain generate a novel computational approach for investigating how tradeoffs influence adaptive dynamics. Specifically the authors use a Markov process to generate maps of all possible evolutionary trajectories under different environmental conditions. The approach differs from previous approaches by allowing larger mutational steps and maintenance of two genotypes. The primary findings are that 1) there are many possible evolutionary outcomes even in simple environments 2) evolutionary outcomes cannot be predicted from environmental conditions and 3) limiting mutation size tends to make evolutionary dynamics more repeatable, but increases the number of steps to an evolutionarily stable endpoint.

The authors provide a large amount of data in this densely packed manuscript. The richness of evolutionary dynamics is interesting. My primary concern is that the author's do so little to connect their work to the field more broadly. For example, how do any of the computational results compare to wet lab observations?

We have made substantial changes to better connect our work to existing knowledge, including statements about the relevant laboratory and field context for the applicable computational results as they appear in the text. In addition, we have entirely re-written and extended the Discussion to emphasise how our results and predictions are relevant to real microbial communities, experimental observations, and to the field's current state with regards to theory.

More specifically, we have included comments and the appropriate references at:

- (i) Lines 103 & 518: for the premise for the phenomenological trade-offs we investigate, which is founded on realistic experimental observations;
- (ii) Lines 128 & 441: for frequency-dependent effects, such as the reversal of the invasibility relationship between a pair of phenotypes in the presence of co-residents, which have been verified both in laboratory evolution experiments and in experimental models of the human gastrointestinal tract, where they are pervasive;
- (iii) Lines 219 & 452: showing that the multi-stationary evolutionary outcomes we observe are connected to the alternative community states detected in planktonic ecosystems and in the gastrointestinal tract;

- (iv) Lines 226 & 454: remarking that our evolutionary cycling outcomes are similar to the ecological cycling demonstrated in a microbial food web, although we note that the two phenomena have an important difference in timescales, and are related to results on evolutionary chaos;
- (v) Line 479: noting the similarity between the topological features that we extract from networks of mutational paths to the statistical indicators of predictive early-warning systems and highlighting that a short-coming of these early-warning models is that only a single ecological transition is considered, whereas our models predict qualitative community changes over longer time-scales involving multiple transitions;
- (vi) Lines 34 & 457: connecting the repeatability of adaptation dynamics that we study to evolution experiments that demonstrate how replicate evolving populations have both unique and common adaptive changes and further predicting that repeatability in adaptation can be controlled through choosing the experimental conditions in chemostats;
- (vii) Lines 482 & 489: framing our methodology in a broader theoretical context, including a new paragraph to connect our network models to instances where networks have been used previously in ecology and evolution;
- (viii) Line 512: describing an evolution experiment to validate our theoretical results.

Further, it is important to highlight that the new method recapitulates previous results (i.e. supplementary figures 4 and 5 should be discussed more).

In the paragraph starting on line 151, we now draw attention to Supplementary Figures 4 & 5 to indicate that as well as extending existing theory we recover previous theoretical results. We also include a discussion of the effects of frequency-dependent invasion fitness described in Supplementary Figure 4 in a new paragraph starting on line 142 and in Fig. 3B.

A vast array of network analysis is carried out, but this analysis is weakly connected to biological understanding.

We find this statement too harsh and, indeed, reviewer 2 refers to the same analysis as ‘an exciting direction for researchers’.

In particular:

- (i) In the section now entitled ‘Mutational path properties identify evolutionary outcomes’, we discuss properties of the mutational paths that all have a direct biological interpretation in terms of the number of successful mutations required to reach the long-term outcome. A more complex example is the minimum cut size, which is a measure of the extent of evolutionary bottle-necks in the mutational process.
- (ii) Our analysis of predicting the evolutionary outcome from the environment (Figure 4) is clearly a biologically important question and, as we show, has relevance for both the complexity and stability of microbial communities.
- (iii) Figure 5 uses network analysis to demonstrate that different evolutionary outcomes have characteristic dynamics of adaptation, potentially a fundamental property of evolution.
- (iv) Figure 6 addresses predicting evolutionary outcomes from the partially sampled networks of mutational paths found in experiments. The network centralities used may be abstract quantities, but we argue that their biological relevance is not important here where the focus is instead to find indicators that work. Such indicators are still useful even if they themselves are not biologically interpretable.

Minor suggestions:

- 1) **The data in figure 5C is difficult to see. Perhaps darker colors could be used?**

We apologize and have now fixed a bug in our figure-generating code that did not correctly render some lines. The panels in Figure 6 are also easier to see now.

2) I found the direction of the y-axis in figure 6B confusing, but this is a personal preference.

We transposed the horizontal and vertical axes in Figure 6B and 6C to make all horizontal axes in Figure 6 represent the degree of network completion.

3) It would be nice to see some discussion of how mutational constraints might influence changes in phenotype over evolutionary time. I am not suggesting more analyses, just discussion of the fact that the phenotypic changes that are being modeled would actually be driven by genetic mutations. Some mutations may be more common than others, and some phenotypes may be mutationally inaccessible (reversions may be particularly unlikely). Additionally, evolvability/robustness of organisms might change as a result of adaptation.

We thank the reviewer for pointing out this implicit assumption in our model, and we have now added some sentences to the discussion highlighting how we include such constraints and the changes we expect if the constraints are altered (lines 504-511).

4) In the last section of the results it would be useful to see more discussion of how incomplete networks relate to a single adaptive trajectory. A single adaptive trajectory seems different than partial mapping of the complete network of all possible evolutionary trajectories. I suspect that one would get much less information from a single adaptive trajectory than is being considered in the partial networks.

We agree that this point is important and have now added a comment in line 378 to indicate that a single incomplete trajectory is unlikely to contain enough information to enable reliable prediction. Nevertheless, the experimental scenario we describe in line 369 is a common one — multiple replicate evolution experiments running in parallel where each reveals an incomplete adaptive trajectory — and is closer to how we envisioned networks of mutational path networks might be constructed.

Response to reviewer 2

In the manuscript, authors Josephides and Swain propose a simple model on the competition and the accompanying adaptive evolution of microbial communities, and perform *in silico* experiments on it to shed light on its dynamics and emerging patterns. They hierarchically separate the evolutionary outcomes of this model into seven broad categories defined in terms of the phenotypes of the constituent populations, and study how these outcomes depend on a) environmental parameters, b) the structure of the network of mutational paths that represent the dynamics of the adaptation process. Taking one further step in the latter direction, they make the argument that in more complex models that involve more environmental parameters and more diverse populations, the computational burden might prevent the comprehensive simulation of the entire system (i.e. the determination of all possible mutational paths) and therefore predicting those outcomes on incomplete networks becomes of essence. They demonstrate that this is possible using simple centrality measures.

In general, I found the paper well-written and the subject matter compelling. The suggested model is simplistic but the resulting dynamics are rich, which lends itself well to a system-wide treatment by the use of graph theoretical tools. Networks are increasingly being used to represent associations between microbial communities, and the extension of the same approach to networks representing temporal steps of evolution presents an exciting direction for researchers. That said, the paper has considerable room for improvement to make it more accessible to the general audience. Below are my specific and general comments.

1) As a general comment, it may be a little hard at times to follow the text because some eco-evolutionary concepts and terms are used without much introduction. As it is now, it looks more like it is aimed at a topic-specific journal than the broad readership of Nature Communications. It could benefit from having some basic concepts, even if they're completely fundamental to readers in the field, defined in one sentence in order to prevent it from sounding like jargon. For example, there could be a brief definition of chemostat. It's simple enough a concept — just needs to be described shortly. The same goes for “import capacity” on line 68.

We now explain concepts and keywords that may be unfamiliar to a general audience, such as the chemostat (line 63), the weak-mutation limit (line 116), and frequency-dependent fitness (line 142).

Along the same lines, the Fig. 1C paragraph (lines 82-86) can be made much more “user-friendly”. The use of terminology like “substrate specific yield” (instead of mass, I guess?), etc. makes it hard to follow whereas the basic premise is quite simple. Cells eat u and v at rates proportional to u and v concentrations and divide once a certain mass is reached. I think all of the above can simply be circumvented by defining “import rate”, “metabolic rate”, “yield”, “growth state” etc. in simple terms once and for all, in Figure 1's caption.

To improve readability of this section, we divided the section describing ecology into two subsections: the first describes the chemostat growth environment (line 62); the second describes only the metabolic specialization trade-off (line 90). In the first subsection, we simplified our description of the microbial growth cycle, particularly with regard to progression through the space of growth states and to substrate import, metabolism, and yield. A new paragraph (line 85) summarizes the nine environmental parameters that describe the chemostat model. To reduce the amount of information required to understand the relevant aspects of ecology, we moved the paragraph describing mutant invasion to the next section (line 111), since it is more appropriately read as an introduction to mutation-limited adaptation.

2) The waiting time between state transitions is ignored in the paper. Can the authors comment on whether or not the time elapsed between transitions has any implications in terms of the evolutionary time scale?

We omit time between state transitions in our models because the assumption that ecological and evolutionary timescales are perfectly separated complicates the calculation and interpretation of waiting times. Our models therefore do not quantify the evolutionary time scale but rather only work with the sequence of mutations. We add a comment to clarify this point in line 166 and explain in the Discussion (line 497) that relaxing the separation of the two timescales requires the development of a general stochastic theory of birth, death, and mutation.

3) Since it's a frequently used parameter in the subsequent analyses, it would be helpful to the reader to include a short biological interpretation of the maximum mutation size ΔS_{\max} with a few words, alongside its formal definition in line 125.

We give a biological interpretation for maximum mutation size with a simple example, showing how one specialist phenotype, but not the other, can be generated from a generalist phenotype, in line 178.

4) Could the authors make the explanation of truncation a little clearer? What exactly is meant by recursion?

We re-worded the description of the dynamic programming algorithm in line 133. We explain that sub-trees below two identical nodes in the invasion tree are the same; therefore, we terminate simulations when we encounter a node that has been handled elsewhere in the tree to avoid repeating the same computations. As a corollary, when the sub-tree below a parent node contains the parent node again, extending the sub-tree would infinitely repeat the sub-tree — that is what we mean by recursion, which we avoid by terminating the simulation paths.

5) In line 146, what exactly are environmental parameter sets? I understand that it's

10,000 realizations with different parameters but it would be helpful to reiterate which environmental parameters were changed.

We address this ambiguity with the new paragraph that collects and summarizes the nine environmental parameters (see point 2) and we change the wording in line 201.

6) In Fig. 3, four out of 10,000 environmental parameter sets are shown. How representative are these? The authors show the statistics of these to some extent in the following sections but how can one make sure that the four types are the only patterns of interest? Is there a way to quantify that?

The example networks in Figure 3 show a sample of interesting properties of adaptation that can be visualized graphically. They are not meant to be representative of all possible patterns that we observed. Indeed, we are unsure about how to appropriately define a network pattern in this context and instead the statistical analyses we undertake in Figure 5 addresses this need to quantify and compare networks. We added a relevant statement in line 236.

7) In lines 181-182, the authors say no clear pattern emerged in Fig. 4A. What was the hypothesis, what kind of an outcome was expected? In other words, how should each of these plots be interpreted to see any “pattern”? The first two panels are (dilution rate and influx) are discussed but what about the interpretation of the last three panels? Also, how are the dotted lines in the second panel of Fig. 4A determined? Are they a guide to the eye or quantified?

We re-wrote and re-structured the section on association of environmental parameters to evolutionary outcomes. The section is now split into two subsections: one for univariate and one for multivariate analysis.

Starting with the univariate analysis, we were hoping that certain types of evolutionary outcome would correspond with certain ranges of the values of the parameter. We now explain how the map from parameters to outcomes is not robust because, with one exception, we did not find parameter values that exclusively led to a single outcome. We extend our discussion to describe how one of the outcomes (multi-stationary) can only be found at certain combinations of substrate yields (although other evolutionary outcomes can be found at these yields too). We also explain how substrate metabolic rates and maximal import rates did not have a large effect, by themselves, on the probabilities with which outcomes emerge. We noted that the dotted lines in the first and second panel are drawn as guides.

The new second subsection describes our multi-variate analysis, the predictive model, and shortest-distance calculations. The lack of robustness in the map from environmental conditions to evolutionary outcomes remains even when multiple (all) parameters are taken into account.

8) Supplementary Fig. 9: How exactly is it decided that it’s poor predictive performance (not reliable) with the given recall rates? They look high to me. How is the mean 0.78 when the lowest recall is 0.83? It may be something obvious that I cannot see, but what was the criterion here? It would help if this were clarified.

We explain in the caption of Supplementary Figure 9 that the hierarchical model’s performance at predicting an outcome is the product of the individual classifiers (nodes) in the hierarchy. While the individual classifiers’ performance appears strong, taking the product of many imperfect classifiers ultimately reduces the overall performance.

9) It’s important to note that Supp. Fig. 10 is not symmetric, i.e. distinguishes between the “from” state and the “to” state. The same applies to Fig. 4C. In fact, both figures can be made easier to understand, especially Fig. 4C, which is a main figure. It’s confusing to have the labels in the same order in the left and right panels. It should simply say distance from monomorphic specialist to other in the left, and other to monomorphic in the right.

We appreciate the subtlety in interpreting these figure. We added a note in the caption of Supple-

mentary Figure 10 to emphasize the asymmetry to the reader, and also emphasized the asymmetry in the text (see point 11 below). We removed the possibly confusing text labels in Figure 4C.

10) On a related note, it's not entirely clear to me why Fig. 4C is not symmetric if it's based on distance. Can the authors clarify the rationale behind this asymmetry? It makes sense that it's much more likely to transition from a diverse population to a specialized one than the other way around, but how does that translate to what we see in the Euclidean distance distributions? A simple interpretation would help the reader a lot here.

The asymmetry is important and arises due to differences in cluster size and shape. We give a simple analogy (line 288) of two dimensional clusters where, even for clusters of the same size, differences in shape can lead to asymmetries in the distribution of shortest distances required to change one outcome to the other.

11) From Fig. 5B, it looks to me as if it's not only dimorphic specialists but also multiple recurrent and monomorphic specialist that don't increase monotonically but rather plateau early on and stay the same with increasing mutation size. The authors, however, say that entropy typically increases with mutation size. That seems to apply only to orange (one generalist), green (two generalists) and hybrid (to some extent).

We amended our wording to be more clear that the monotonicity in path entropy is not strict: i.e. entropy can increase or not change, but it does not usually decrease (paragraph starting on line 318). We expanded on the differences in the 'rate' of plateauing, writing that these might be explained by differences in the probability that mutational paths with large-effect mutations are followed, which might in turn be explained through variation in the importance of frequency-dependent effects on invasion fitness (lines 326-331).

12) In my opinion, Fig. 5A does a better job at simply explaining how to read the mutational path networks (i.e. starts from a monomorphic state, ends at a recurrent state etc.). This explanation should be carried over to Fig. 3B-E as well.

Thank you for this suggestion. We now show a couple of example mutational paths for each network in Figure 3B-E and label the resident populations in selected vertices to highlight notable aspects of adaptation, such as the reversal of invasibility relationships as a consequence of frequency-dependent effects (figure caption).

13) Line 247: Larger mutation size leads to fewer bottlenecks. Could the authors shortly explain what may be reason like they did in the same paragraph for the path lengths?

We now describe one way in which bottle-necks can decrease when the maximum mutation size increases in line 347, which we explain in terms of the increasing number of outgoing edges (possible community transitions) in the network.

14) Supplementary Fig. 11 needs a much more informative caption. The mean and variance of each measure for each network is given but overall the figure doesn't convey to the reader how these centrality measures differ qualitatively from each other. It says, "vertices are scaled..." to mean the vertex size, this should be clarified. At minimum, it should be made clear that, simply, the larger nodes, be it round or square, have higher centrality of the respective kind. Although an even better approach would be to give specific network instances that emphasize each centrality type, e.g. a bottleneck node for high betweenness centrality.

We extended our discussion of network centralities in the supplementary material and clarified what we mean by 'scaling' vertices in the figure caption. We include a new Supplementary Table to describe the six network centralities that we use and added appropriate references. In this table, we also refer to examples from the networks in Supplementary Figure 11 to explain why some vertices have high or low centrality values.

15) Line 287: "Most informative centrality measures converged gradually?" What is

meant by most informative, simply ones with the highest mutual information? In any case, the authors should try to explain intuitively why in-degree of HITS hub are considered more informative, and what it means for them to converge gradually? Does it mean they are more crucial as a centrality measure to predict true recurrent states? If so, this should be stated explicitly to help the reader understand this section better.

We re-wrote the paragraph (line 395) on characterizing centrality features in terms of their mutual information and error (convergence). We clarify what we mean by informative measures: those with higher mutual information. We also clarified what we mean by measures that converge early: those with a small error in the value of the centrality statistic between incomplete and complete forms of the network.

Interpreting why some centrality measures are more informative than others with regards to evolutionary outcomes is challenging. Unlike properties of the mutational paths, which do have biological interpretations, network centrality measures are abstract properties and their mappings to ecology or evolution are not (yet) clear. We added a statement to discuss this difficulty in line 388. We note though that a straightforward interpretation is not necessary to enable accurate prediction.

16) As a final suggestion for a discussion point, I think the authors could recognize some recent efforts on network methods used in microbial evolution modeling such as: Faust, K., & Raes, J. (2012). Microbial interactions: from networks to models. Nature Reviews Microbiology, 10(8), 538-550. I think it's important to comment on the current divide between simple computational models and real-world microbial communities and how current methods can be improved specifically in this direction to align the two worlds.

We extensively re-wrote our Discussion section to address this and other comments. We now acknowledge other uses of network models in ecology and evolution and explain how our approach differs from those studies (line 482). We address the divide between theory and real microbial communities by discussing: first, the necessary theoretical improvements to our and similar models (paragraph starting on line 499) toward biological realism; and second, suggesting a set of experiments that could validate our theoretical predictions (paragraph starting on line 512).

Minor issues and typos:

- 1) Typo in Figure 2 caption, line 3: “phenotypes values”.
- 2) In Supplementary Figs. 1 and 2, μ and m_v are not defined.
- 3) What's the circle next to “truncated node” in Fig. 2? I believe it means darker gray nodes are truncated nodes, this should either be made clear or it should be removed.
- 4) Line 153: A (insert: single) recurrent state may be a dimorphic or...
- 5) Fig. 4B says both “normalized Euclidean distance” on the color bar and “pairwise standardized parameter distance” on the right hand side. Are they not the same? It would be better to standardize the notation.
- 6) Line 182, “Using a nearest-neighbors algorithm...” The authors cite the original paper for this algorithm in the methods section so there should be a reference here to the Methods section.
- 7) Supplementary Fig. 10: I am assuming Euclidean distances, i.e. similarities, are meant by “normalized shortest distances”. If so, this should be made clear so as not to confuse the reader since the term “shortest distance” has a clear connotation in network theory. The shortest distances mentioned here could be confused with the shortest paths or geodesics in the network.
- 8) Figure 2 caption, line 3: “phenotypes values”.

We thank the reviewer for pointing out these errors and have corrected all as well as improving the labels in Figure 2A & 4B, citing a reference for the nearest-neighbor algorithm in Methods, and clarifying the meaning of ‘shortest distance’ in Supplementary Figure 10 (caption).

REVIEWERS' COMMENTS:

Reviewer #1 (Remarks to the Author):

The authors have adequately addressed my concerns.
In line 302 "evolutionarily" should be "evolutionary".

Reviewer #2 (Remarks to the Author):

The authors have diligently addressed all concerns I raised in the first review in a satisfying manner and have thus made the text clearer to the reader. The presentation of the manuscript is now significantly improved.